# Southern Ocean warming and Antarctic ice shelf melting in conditions plausible by late 23[rd] century in a high-end scenario

Pierre Mathiot[1] and Nicolas C. Jourdain[1]

[1]Univ. Grenoble Alpes/CNRS/IRD/G-INP/INRAE, Institut des Geosciences de l'Environnement, Grenoble, France.

**Correspondence:** Pierre Mathiot (pierre.mathiot@univ-grenoble-alpes.fr)

**Abstract.**

How much Antarctic ice shelf basal melt rates can increase in response to global warming remains an open question. Here we describe the response of the Southern Ocean and ice shelf cavities to an abrupt change to high-end atmospheric conditions plausible by the late 23[rd] century under the SSP5-8.5 scenario. To achieve this objective, we first present and evaluate a new 0.25° global configuration of the NEMO ocean and sea ice model. Our present-day simulations demonstrate good agreement with observational data for key variables such as temperature, salinity, and ice shelf melt rates, despite remaining difficulties to simulate the interannual variability in the Amundsen Sea. The ocean response to the high-end atmospheric perturbation includes a strengthening and extension of the Ross and Weddell gyres and a quasi-disappearance of sea ice, with subsequent decrease in production of High Salinity Shelf Water and increased intrusion of warmer water onto the continental shelves favored by changes in baroclinic currents at the shelf break. We propose to classify the perturbed continental shelf as "warm–fresh shelf". This induces a substantial increase in ice shelf basal melt rates, particularly in the coldest seas, with a total basal mass loss rising from 1,180 to 15,700 Gt yr$^{-1}$ and an Antarctica averaged melt rate increasing from 0.8 m yr$^{-1}$ to 10.6 m yr$^{-1}$. In the perturbed simulation, most ice shelves around Antarctica experience conditions that are currently found in the Amundsen Sea, while the Amundsen Sea warms by 2°C. These idealised projections can be used as a base to calibrate basal melt parameterisations used in long-term ice sheet projections.

## 1 Introduction

Most future projections of the Antarctic contribution to sea level rise have so far relied on ice sheet models in which ice shelf basal melt was parametrised from the changing ocean characteristics of global climate simulations (e.g., Cornford et al., 2015; Seroussi et al., 2020; Levermann et al., 2020; DeConto et al., 2021; Payne et al., 2021). Such parametrisations calculate ice shelf basal melt rates from the ocean properties on the continental shelf and do not explicitly represent the ocean circulation and mixing in ice shelf cavities, including the crucial interactions with bathymetric features and tides (Burgard et al., 2022). They are directly fed by the outputs of global climate simulations that are highly biased near Antarctica (Little and Urban, 2016; Barthel et al., 2020), partly due to their coarse resolution (van Westen and Dijkstra, 2021) and to the absence of feedbacks between glacial meltwater and the climate system (Donat-Magnin et al., 2017; Bronselaer et al., 2018; Sadai et al., 2020; Li et al., 2023). For these reasons, a number of modelling centers are currently incorporating interactive Antarctic Ice Sheet

models into their climate models (e.g., Smith et al., 2021; Pelletier et al., 2022). For this, the ocean components of climate models need to represent the ocean circulation beneath ice shelves.

Simulating the ocean properties and ice shelf melting at a circum-Antarctic scale is difficult because it is highly sensitive to the ocean and sea ice model settings. For example, some model settings can make the Ronne-Filchner cavity tip into a warm state (Comeau et al., 2022) or the Amundsen Sea switch to a cold state (Naughten et al., 2018b; Smith et al., 2021) under present-day or pre-industrial conditions. Such biases raise concerns on the validity of ocean–ice-sheet projections in some important regions of Antarctica (Timmermann and Hellmer, 2013; Naughten et al., 2018a). In this paper, we present a new configuration of the NEMO (Nucleus for European Modelling of the Ocean NEMO System Team, 2019) ocean–sea-ice–ice-shelf model at 0.25° resolution that represents reasonably well the seas and ice shelf cavities around Antarctica.

In addition to a strong sensitivity to model settings, the present-day ice shelf melt rates are highly sensitive to present-day biases in the driving climate models, which are particularly important around Antarctica (Agosta et al., 2015; Barthel et al., 2020; Purich and England, 2021). This again raises concerns on the validity of ocean–ice-shelf projections starting from highly biased present-day conditions. Given that the climate model biases are largely stationary even under strong climate changes (Krinner and Flanner, 2018), it can be relevant to use bias correction methods (Krinner et al., 2020) or to constrain projections by anomalies with respect to present day (Donat-Magnin et al., 2021; Jourdain et al., 2022). In this paper, we use an anomaly method to explore Southern Ocean warming and Antarctic ice shelf melting in a plausible late 23$^{rd}$ century, under a high-end (extremely unlikely) scenario. Our projection method is highly idealised but it can be useful for theoretical studies on ocean tipping points, for a first investigation on circum-Antarctic melt rates in a much warmer climate, and to calibrate ice shelf basal melt parameterisations used for high-end long-term ice sheet projections.

## 2 Ocean–sea-ice–ice-shelf simulations

### 2.1 Model

The ocean model used in this study is based on NEMO version 4.0.4, which represents the ocean dynamics and physics (NEMO-OPA, NEMO System Team, 2019) and the sea ice dynamics and thermodynamics (SI$^3$, NEMO Sea Ice Working Group, 2019). The migration from 4.0.3 to 4.0.4 version contained a critical bug on the distribution of solar and non-solar fluxes over sea-ice covered areas but this was fixed in the version used in the version used in this study (complete description of the bug on https://forge.ipsl.jussieu.fr/nemo/ticket/2626). The configuration used in this study is the so-called eORCA025, a quasi-isotropic global tripolar grid with a 0.25° nominal resolution which was extended southward to represent Antarctic under-ice-shelf seas (Mathiot et al., 2017; Storkey et al., 2018). A nonlinear free surface using the variable volume layer formulation is applied (Adcroft and Campin, 2004). The vertical discretisation is made on 121 levels with a resolution of 1 m at the surface, 20-30 m between 100 and 1000 m depth, and up to 200 m at 5000 m depth. This is a much finer and more uniform vertical resolution in the deepest part of the ice shelf cavities than the 75 levels previously used in the NEMO community (e.g., Mathiot et al., 2017; Hutchinson et al., 2023; Smith et al., 2021), with 30 m at 1000 m instead of 100 m. Partial steps (Barnier et al., 2006) are used to represent the actual bathymetry and ice shelf draft.

Bathymetry and ice shelf draft are similar to the ones used in Storkey et al. (2018) and updated toward Bedmachine Antarctica v2 on the Antarctic continental shelf (Morlighem et al., 2020; Morlighem, 2020). Because of its effect on sea ice and water masses (mean state and variability) in the Amundsen Sea (Bett et al., 2020), the line of icebergs grounded on Bear Ridge has been added as land points blocking the advection of sea ice. After preliminary tests, the Getz ice shelf draft was artificially thinned by 200 m (keeping the grounding line unchanged) in order to compensate a longstanding bias in the thermocline depth (previously reported by Mathiot et al., 2017). The later was driving very excessive release of meltwater, which was strongly deteriorating the mean state of the Ross Sea (a connection previously described in Nakayama et al., 2020). More details on the impact of such correction are provided in section 3.3.

The horizontal and vertical advection of tracers is done using fourth and second order Flux Corrected Transport scheme (Zalesak, 2012), respectively. A polynomial approximation of the TEOS10 equation of state is used (Roquet et al., 2015). A parameterisation of adiabatic eddy mixing (Gent and Mcwilliams, 1990) is activated where the Rossby radius is smaller than 2 times the model grid resolution, with a coefficient of $150\,\mathrm{m^2\,s^{-1}}$. Internal wave mixing is parameterised following de Lavergne et al. (2016).

A free-slip lateral boundary condition on momentum is applied with no slip condition applied locally at Bering Strait, in the whole Mediterranean sea, along the West Greenland coast and around the south Shetland, Elephant and south Orkney islands (at the Northern end of the Antarctic Peninsula). This technique is a crude method to take into account the locally complex sub-grid scale bathymetry, and it affects water mass properties as explained in section 3.2. A quadratic bottom friction is used with a drag coefficient of $10^{-3}$ and increased values in the Torres, Denmark, and Bab-el-Mandeb straits, as well as around the south Shetland, Elephant and south Orkney islands. A 3d damping toward WOA2018 (World Ocean Atlas 2018 Locarnini et al., 2019; Zweng et al., 2019) is done in the Red sea and Persic gulf (time scale of 180 days), as well as strong restoring downstream of the Gibraltar (600-1300m), Bab-el-Mandeb and Ormuz straits (time scale of 6 days). All the aforementioned changes (except changes near Antarctic Peninsula) in slip condition, bottom friction and 3d damping are similar to the ones used in Molines et al. (2007).

Other modelling choices such as momentum advection, lateral diffusion of momentum and tracer, vertical mixing, convection, double diffusion, and bottom boundary layer are set as in Storkey et al. (2018).

$SI^3$ is a multi-layer and multi-category sea ice model. In this study, we use the default setting of $SI^3$ provided by the NEMO distribution except the ones described hereafter. We use the elastic-viscous-plastic rheology described in Bouillon et al. (2013). The ocean–sea-ice drag coefficient is set to $5 \times 10^{-3}$. Snow thermal conductivity is set to $0.35\,\mathrm{W\,m^{-1}\,K^{-1}}$ and maximum sea ice fraction is 0.95 as in Boucher et al. (2020) to account for non-resolved leads and polynyas. Such a low maximum sea ice fraction is required to maintain realistic dense shelf water properties on cold shelves (too fresh otherwise) and CDW properties on warm shelves (too warm otherwise) in our experiments. A sea ice monthly climatology from a global GO6 simulation (Storkey et al., 2018) forced by the JRA55-do atmospheric reanalysis (Tsujino et al., 2018) over the period 1980 to 2004 is used as initial conditions for sea ice concentration and thickness.

Iceberg melt is computed on-line using the Lagrangian iceberg module implemented in NEMO (Marsh et al., 2015; Merino et al., 2016). Icebergs are categorised in the same ten classes as Gladstone et al. (2001). The historical iceberg distribution

used in NEMO (Marsh et al., 2015; Merino et al., 2016) being biased toward small icebergs (Stern et al., 2016), we use the
mass-weighted distribution proposed by Stern et al. (2016). Its distribution follows the -3/2 power law iceberg-size distribution
observed by Tournadre et al. (2016). The total calving rate of individual ice shelves is derived from Rignot et al. (2013) who
assumed steady ice shelf fronts. As we have no information on the geographical distribution of calving for a given ice shelf,
the local calving rate of each ocean cell along the front of an ice shelf is defined randomly at the beginning of the simulation,
in a way that preserves the total amount of calving per ice shelf. The calving rate is kept unchanged throughout the simulation.

Ocean circulation and basal melt in ice shelf cavities is derived from the NEMO module described in Mathiot et al. (2017).
The calculation of ice shelf melt rates follows the standard three-equation parameterization (Holland and Jenkins, 1999; Jenkins
et al., 2001) with a velocity dependant formulation (Jenkins et al., 2010) as described in Asay-Davis et al. (2016). Heat ($\Gamma_T$)
and salt ($\Gamma_S$) exchange coefficients are respectively $1.4 \times 10^{-2}$ and $4.0 \times 10^{-4}$, while the top drag coefficient ($C_d$) is set to
$2.5 \times 10^{-3}$, which gives a thermal Stanton number ($St$) of $0.7 \times 10^{-3}$ as in Jourdain et al. (2017), Hausmann et al. (2020) and
Bull et al. (2021). The ocean conservative temperature, absolute salinity, and velocity used in the three-equation parametrization
are averaged over a top boundary layer of constant 20-m thickness (Losch, 2008; Mathiot et al., 2017). The top background
tidal velocity is derived from CATS2008 (Padman et al., 2008) and applied in the top boundary layer beneath ice shelves
following Jourdain et al. (2019) to increase the ice–ocean turbulent exchange. This tends to increase the heat and salt transfer
velocity and therefore to consume the available heat faster. The ice shelf thickness is constant, so it is assumed that the ice
sheet dynamics instantaneously compensate the melt-induced ice shelf thinning.

In addition to the freshwater flux from iceberg and ice shelf melting, we apply the global river runoff provided by Dai
and Trenberth (2002). Runoff from melting at the surface of the Antarctic Ice sheet is not accounted for as it is currently
negligible compared to other freshwater sources (Agosta et al., 2019). On top of other freshwater fluxes (precipitation, runoff,
...), a common practice in forced ocean models is to use some form of sea surface salinity restoring. This restoring is required
because of the missing atmospheric feedbacks on humidity in forced models (for more details see Griffies et al., 2016). To
make the model sensitivity analysis more robust, this corrective term was diagnosed from sea surface salinity restoring towards
WOA2018 over the period 1999-2018 in a former simulation (the "REALISTIC" simulation described in Burgard et al., 2022)
and applied as an additional climatological monthly freshwater flux in all our simulations.

## 2.2 Experiments

Our present-day simulation is driven by the JRA55-do atmospheric reanalysis (Tsujino et al., 2018) through the CORE bulk
formulae described in Griffies et al. (2009) and Large and Yeager (2004). This simulation is referred to as "REF" and is
initialised in 1979 from the climatological WOA2018 conditions. Our perturbation experiment ("PERT") is an idealised abrupt
change from present day to high-end conditions at the end of the 23$^{\rm rd}$ century. PERT bifurcates from REF in 1999, i.e. after 20
years of spin up under present-day conditions. The same surface fresh water correction flux as in REF is applied to PERT.

To build the perturbed surface forcing, we add an anomaly (2260-2299 minus 1975-2014) to all the present-day fields used to
calculate the surface boundary conditions. The anomaly is extracted from monthly outputs of the IPSL-CM6A-LR projections

(Boucher et al., 2020; Lurton et al., 2020) under the SSP5-8.5 emission scenario (low-probability, high-end anthropogenic emission scenario, Meinshausen et al., 2020).

IPSL-CM6A-LR is one of the few CMIP6 models extending their scenario-based projections to 2300. In present-day conditions, IPSL-CM6-LR is cold biased by a few degrees at the surface of the Antarctic Ice Sheet (Boucher et al., 2020). On the ocean side, bottom water formation on Antarctic shelves is reasonably well represented as well as the presences of the cold and warm shelves in IPSL-CM6 (Heuzé, 2021; Purich and England, 2021). Sea ice extent is within the observational uncertainty in summer and slightly overestimated in winter (Boucher et al., 2020). These elements give confidence that the overall atmospheric forcings of IPSL-CM6-LR can be used to drive an ocean model.

The anomaly is calculated separately for each calendar month, i.e., we apply an anomaly that includes a seasonal cycle. Monthly anomalies are then interpolated between the middle of two consecutive months to avoid discontinuity of the surface boundary conditions. Finally, we cycle the 40-year interannual period (1979-2018) to which the anomaly is applied in order to be able to apply the perturbation over long periods. Our method is expected to correct a part of the CMIP model biases that are largely stationary even under strong climate changes (as shown by Krinner and Flanner, 2018, from preindustrial to 4xCO2). The main caveat of the anomaly method compared to a direct forcing by the IPSL-CM6A-LR projection is that we assume a stationary interannual variability with respect to the mean state (see discussion in Jourdain et al., 2022).

Other aspects of our method make our perturbation very idealised. First of all, by imposing a step change, we neglect the slow component of global ocean warming that is present in continuous simulations from present day to 2300. From this perspective, our perturbation is more similar to the abrupt quadrupling of the atmospheric concentration of carbon dioxide used in the CMIP deck than to a usual scenario-driven projection. Furthermore, we assume that despite large changes in ice shelf basal melting, the ice shelves extent and thickness will remain unchanged until 2300, that iceberg calving rates will remain at their present-day values, and that runoff from ice melting at the surface will remain zero. All these assumptions are unrealistic even for projections to 2100 (Seroussi et al., 2020; Kittel et al., 2021).

We also want to make clear that we use a high-end perturbation even for 2300 conditions. This corresponds to a median global air temperature warming of 9.6°C in 2300 with respect to 1850-1900 (according to the multi-model emulation of Lee et al., 2021). The emission scenario itself (SSP5-8.5) is a low-probability scenario (Hausfather and Peters, 2020). Furthermore, IPSL-CM6A-LR has an equilibrium climate sensitivity (ECS) of 4.6°C (Meehl et al., 2020), which is relatively high given the 66% probability of an ECS below 4.0°C and the 90% probability of an ECS below 5.0°C according to the 6[th] assessment of the Intergovernmental Panel on Climate Change (IPCC, Forster et al., 2021). We nonetheless believe that such an experiment is needed for theoretical studies on ocean tipping points, for a better understanding of circum-Antarctic melt rates in a much warmer climate, and to calibrate ice shelf basal melt parameterisations used for long-term projections.

The main characteristics of the atmospheric perturbations are shown in Fig. 1. They contain strong increase in precipitation, especially along the ice sheet margins, surface air warming far above the global mean warming, especially in austral winter (polar amplification), and strengthening and southward migration of westerlies around Antarctica, particularly in austral summer and fall.

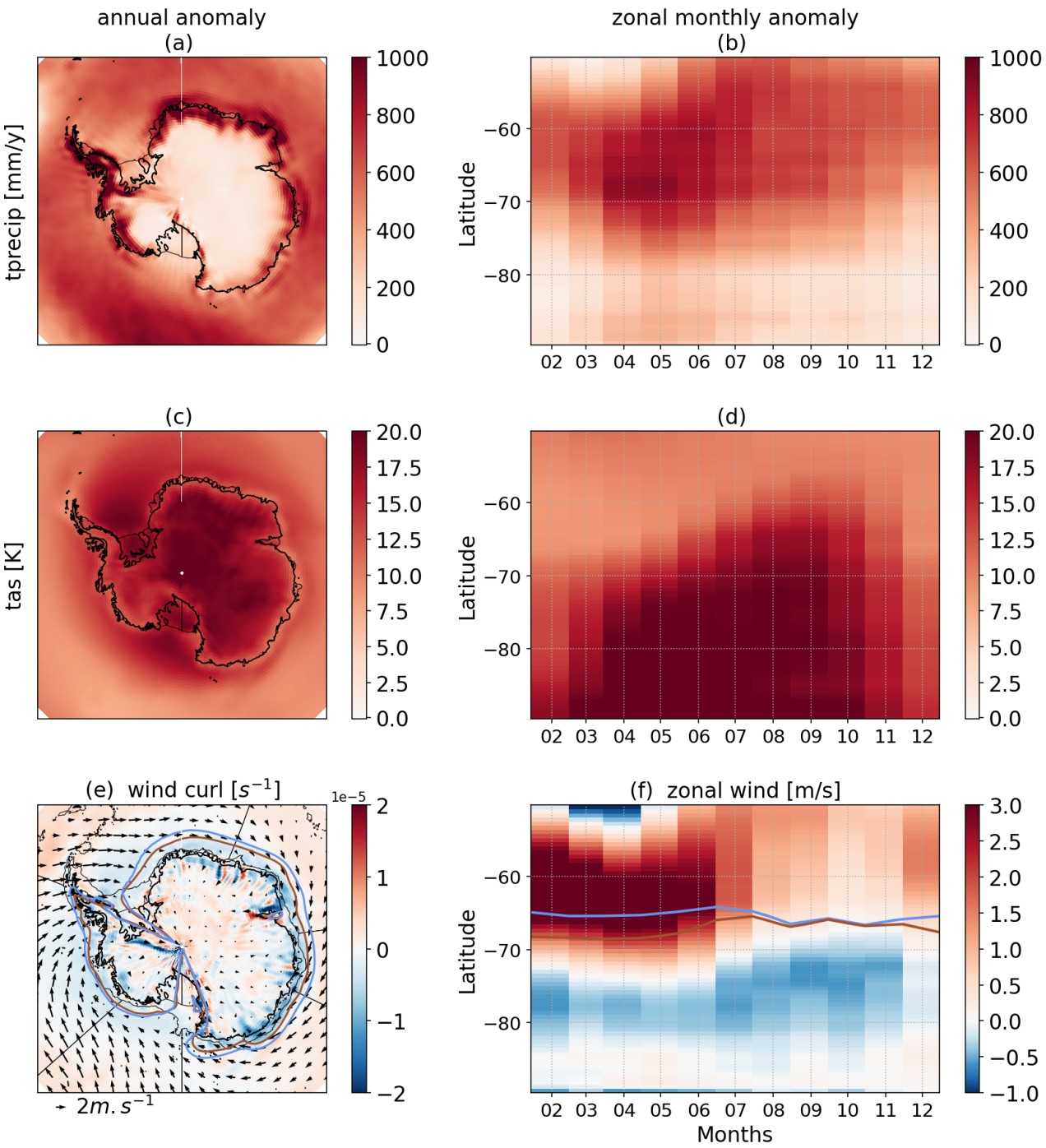

**Figure 1.** Maps of annual mean anomalies (left) and monthly-mean zonal-mean anomalies (right) for (a,b) total precipitation and (c,d) near surface air temperature, (e) near surface wind curl and wind anomalies (arrows) (f) near surface zonal wind . In (e) and (f), the blue line is the position of the zero zonal wind in the JRA reanalysis, and the brown line is the equivalent for the perturbed wind.

# 3 Evaluation of present-day simulations

Simulating realistic properties of the Southern Ocean and Antarctic marginal seas has often been challenging at a 0.25° resolution (e.g., Smith et al., 2021), largely because this resolution is in the grey zone between fully resolved and fully parameterised eddies. The present-day simulation described in this paper is the result of many months of empirical tuning, and it gives a relatively good representation of the ocean-ice properties in the Southern Ocean. We therefore provide an extensive evaluation of this simulation. The results presented here are based on the climatology of the last 10 years of the present-day simulation, i.e., the period spanning from 2009 to 2018.

## 3.1 General circulation

The Antarctic Circumpolar Current (ACC) barotropic transport across the Drake passage is 137 Sv in REF. It compares reasonably well with estimates derived from observations and ocean model reanalyses. Equivalent observational estimates indeed reached 137±8 Sv (Cunningham et al., 2003), 173±11 Sv (Donohue et al., 2016) and 141±13 Sv (Koenig et al., 2014). Ocean reanalyses give an average transport across the Drake Passage of 153±5 Sv in SOSE (Southern Ocean State Estimate, Mazloff et al., 2010), 155 Sv in GLORYS12 (Mercator global ocean reanalysis, Lellouche et al., 2021; Artana et al., 2021), and 152±19 Sv in an ensemble long-term mean transport from nine ocean reanalysis products (Uotila et al., 2019).

The barotropic transport within the Ross and Weddell gyres is reasonably well represented in REF, with 26 Sv and 60 Sv, respectively, as estimated from the maximum barotropic stream function in the southern limb of each gyre (Fig. 2). This is slightly stronger than the equivalent estimates in the SOSE reanalysis (20±5 Sv and 40±8 Sv within the Ross and Weddell gyres in Mazloff et al., 2010). This is nonetheless closer to the observation-based barotropic transport for the Weddell Gyre: 56±8 Sv across the Prime Meridian (Klatt et al., 2005) and 83±22 Sv in the south-eastern limb of the gyre (Reeve et al., 2019).

The shape of these gyres is also realistic (Fig. 2). The Weddell Gyre in REF has a similar shape as the one derived from observations in Reeve et al. (2019, their Fig. 4), although our maximum is located along the prime meridian, i.e., ∼15° westward of their observational estimate. Characterizing the gyre extent by the barotropic stream function at half of its maximum value within the gyre, the Weddell Gyre extends eastward to 45°E in REF vs 30°E in SOSE. Similarly, the Ross gyre seats within [180°E-220°E] in REF vs [160°E-220°E] in SOSE (Mazloff et al., 2010).

## 3.2 Sea ice and water mass properties on the continental shelf

Our REF simulation captures very well the maximum sea ice extent (Fig. 3b), with a September average of 18.3 million km$^2$ in REF vs 18.7 in the satellite product of Meier et al. (2021). REF underestimates the minimum sea ice extent (Fig. 3a), with 2.5 million km$^2$ on average in February vs 3.1 in the satellite product of Meier et al. (2021). This underestimation is mostly due to the missing summer sea ice along the East Antarctic coast (Fig. 3a). A possible explanation for this is the absence of polynyas associated with thick sea ice fastened to grounded icebergs (Nihashi et al., 2017), which would require a specific parameterisation of the sea ice tensile stress (Van Achter et al., 2022), an iceberg grounding scheme, and 2-way icebergs–sea-ice interactions in NEMO.

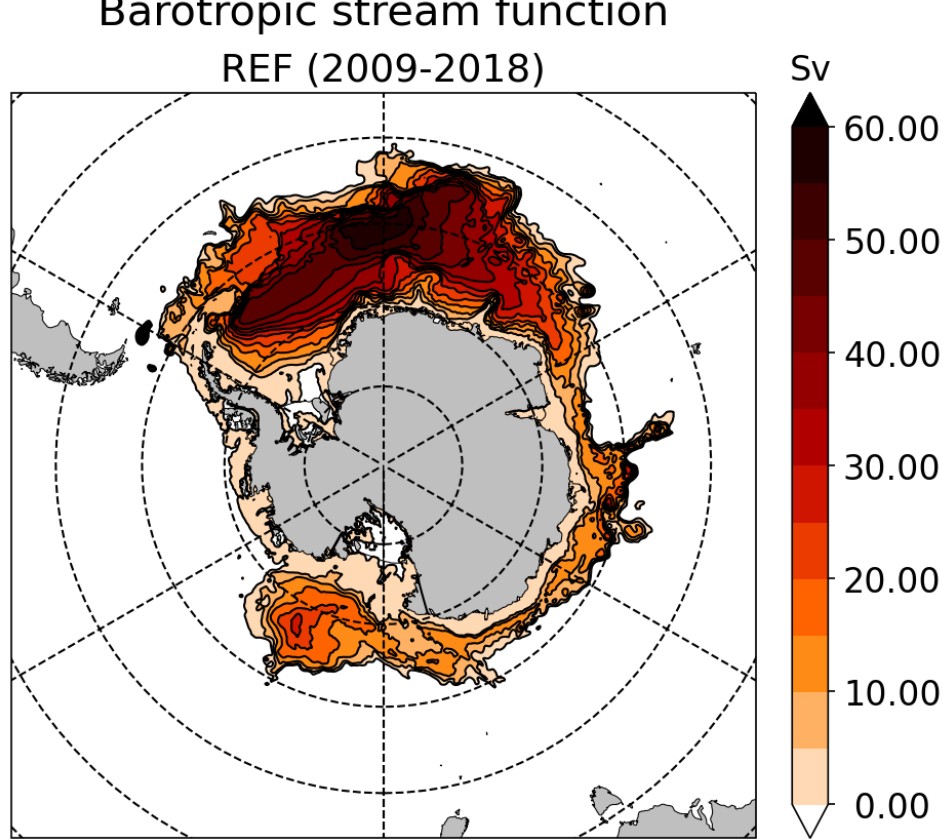

**Figure 2.** Climatological (2009-2018) barotropic stream function ($\Psi$) in REF (contours every 10 Sv). Areas beyond the polar gyres are in white. The zonal and meridional barotropic transports (Sv) between two locations are given by $U = -\Delta_y \Psi$ and $V = \Delta_x \Psi$, i.e. by the signed differences in $\Psi$ between these two points.

The presence of High Salinity Shelf Water (HSSW) is important both because it is a precursor for the Antarctic Bottom Water, which is key for the global thermohaline circulation, and because it controls the circulation in cold ice shelf cavities (e.g., Janout et al., 2021). Our REF simulation produces HSSW in the Ross Ice Shelf and Terra Nova Bay polynyas (western Ross Sea) and in the Ronne polynya (western Weddell Sea) with a reasonable fresh bias of $\sim$0.05 g kg$^{-1}$ (Fig. 4a,b). REF is also slightly too fresh in Prydz Bay, near Amery Ice Shelf, another area known for HSSW production (Herraiz-Borreguero et al., 2015), and in East Antarctica in general, likely due to the aforementioned absence of iceberg-induced polynyas (Fig. 4c).

In terms of bottom temperatures, REF represents weakly modified Circumpolar Deep Water (CDW) in the Bellingshausen and Amundsen Seas, in good agreement with WOA2018 data (Fig. 4c,d). This is an improvement compared to previous circum-Antarctic studies at similar resolution (e.g., Mathiot et al., 2017; Naughten et al., 2018b). As observed, the bottom Weddell and Ross Seas and the adjacent ice shelf cavities are filled with water near the surface freezing point ($-1.9°$C). The simulated bottom temperatures are colder than WOA2018 in the Indian Ocean sector of East Antarctica compared to WOA2018 with

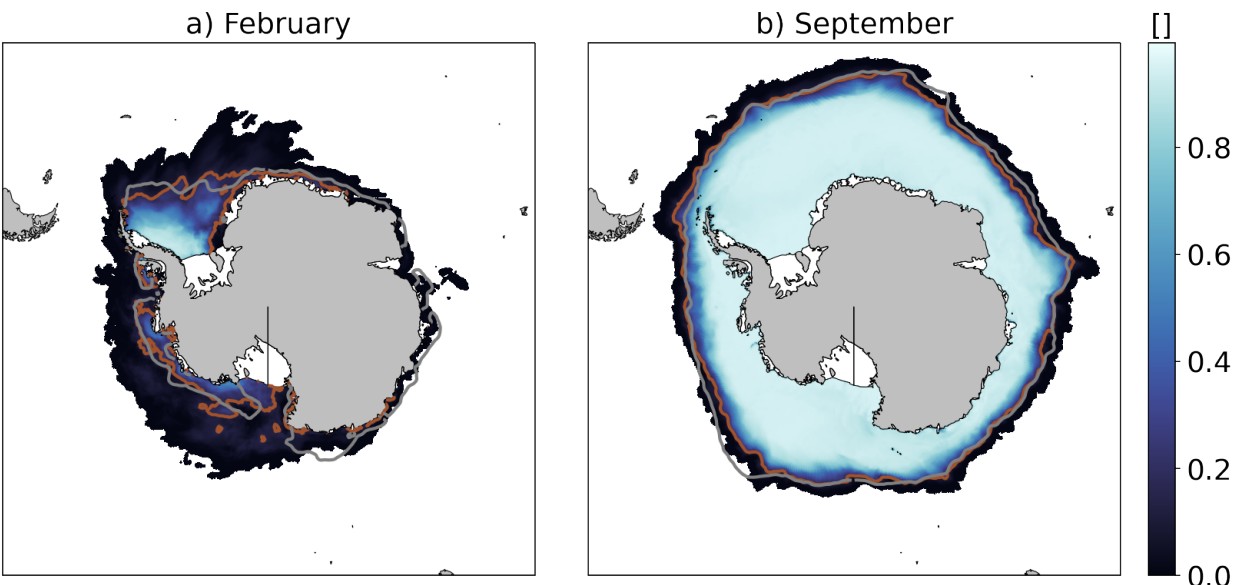

**Figure 3.** Climatological (2009-2018) sea ice concentration in REF in (a) February and (b) September. The grey lines indicate the sea ice extent in the satellite product of Meier et al. (2021), and the brown line shows the equivalent in our REF simulation.

the exception of Prydz Bay/Amery Ice Shelf (Fig. 4f). This comparision should be considered with caution given the sparse observations in this region. Relatively warm water at depth was observed in the vicinity of Totten Ice Shelf and Vincennes Bay, but the presence of cold water was reported at other locations in this sector (Rintoul et al., 2016; Ribeiro et al., 2021; Portela et al., 2022). The north end of the Antarctic Peninsula also exhibits a cold bias in REF. Preliminary analyses during the tuning process suggested that this bias was sensitive to the HSSW properties (worse when HSSW was not dense enough), to the treatment of the bathymetry, and to the lateral slip condition and bottom friction at the tip of Peninsula.

## 3.3 Ice shelf melt

Our ocean model configuration represents 1.48 million $km^2$ of ice shelves, and their simulated rate of basal mass loss is 1182 Gt yr$^{-1}$ (gigaton per year) over 2009-2018, which compares well with the $1325 \pm 235$ Gt yr$^{-1}$ in the 2000s for a surveyed area of 1.55 million $km^2$ in Rignot et al. (2013), and with the $965 \pm 265$ Gt yr$^{-1}$ over 1992-2017 for a surveyed area of 1.54 million $km^2$ in Paolo et al. (2023).

The mean basal mass loss of individual ice shelves is generally in agreement with observational estimates (Fig. 5b). The melt rates are particularly overestimated for Georges VI (a long standing bias in NEMO), while they are significantly underestimated Thwaites. For Thwaites, it should be noticed that we use a recent ice shelf draft in NEMO (Morlighem et al., 2020; Morlighem, 2020) with a significantly reduced area compared to the period covered by Rignot et al. (2013), which logically decreases

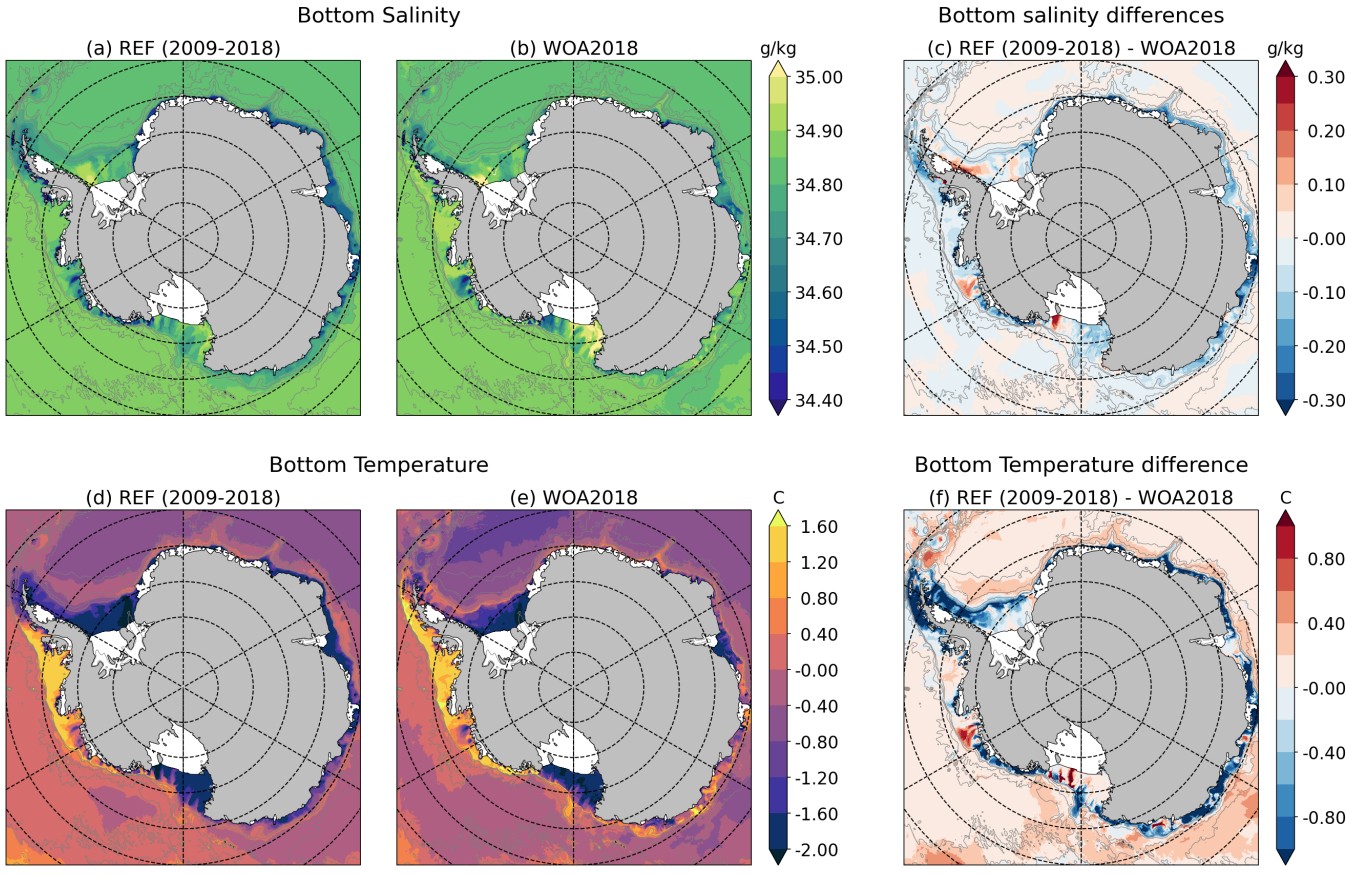

**Figure 4.** Upper panels: climatological bottom practical salinity in (a) REF, (b) WOA2018 (Locarnini et al., 2019; Zweng et al., 2019) and (c) the difference REF-WOA2018. Lower panels: climatological conservative temperature in (d) REF (e) WOA2018 and (f) the difference.

the integrated melt. The total melt underneath Getz was strongly overestimated in preliminary simulations, reaching 400-500
220 Gt yr$^{-1}$ (not shown). By reducing the ice shelf draft of Getz (section 2), we have artificially displaced it into the model cold mixed layer, which gives more realistic melt rates. This empirical correction of the ice shelf draft is nonetheless slightly too strong because it was done prior to the completion of parameter tuning.

The simulated ice shelf melt rate pattern is shown in Fig. 5a. The melt pattern underneath the Filchner-Ronne ice shelf includes large areas of refreezing and maximum melt rates at the front of Ronne Ice Shelf and near the deepest parts of the
225 grounding line, consistently with satellite estimates (Rignot et al., 2013; Moholdt et al., 2014; Adusumilli et al., 2020) and high-resolution simulations (Hausmann et al., 2020). Near zero melt rates are simulated underneath most of Ross Ice shelf, except near Ross Island at the west end of the ice front, as reported by Rignot et al. (2013) and Adusumilli et al. (2020). The warm ice shelves from Getz to Pine Island, in the Amundsen Sea, all experience local melt rates above 10 m yr$^{-1}$ in agreement with the aforementioned satellite estimates. The deepest part of Pine Island shows a high melt area above 20 m yr$^{-1}$, corresponding to

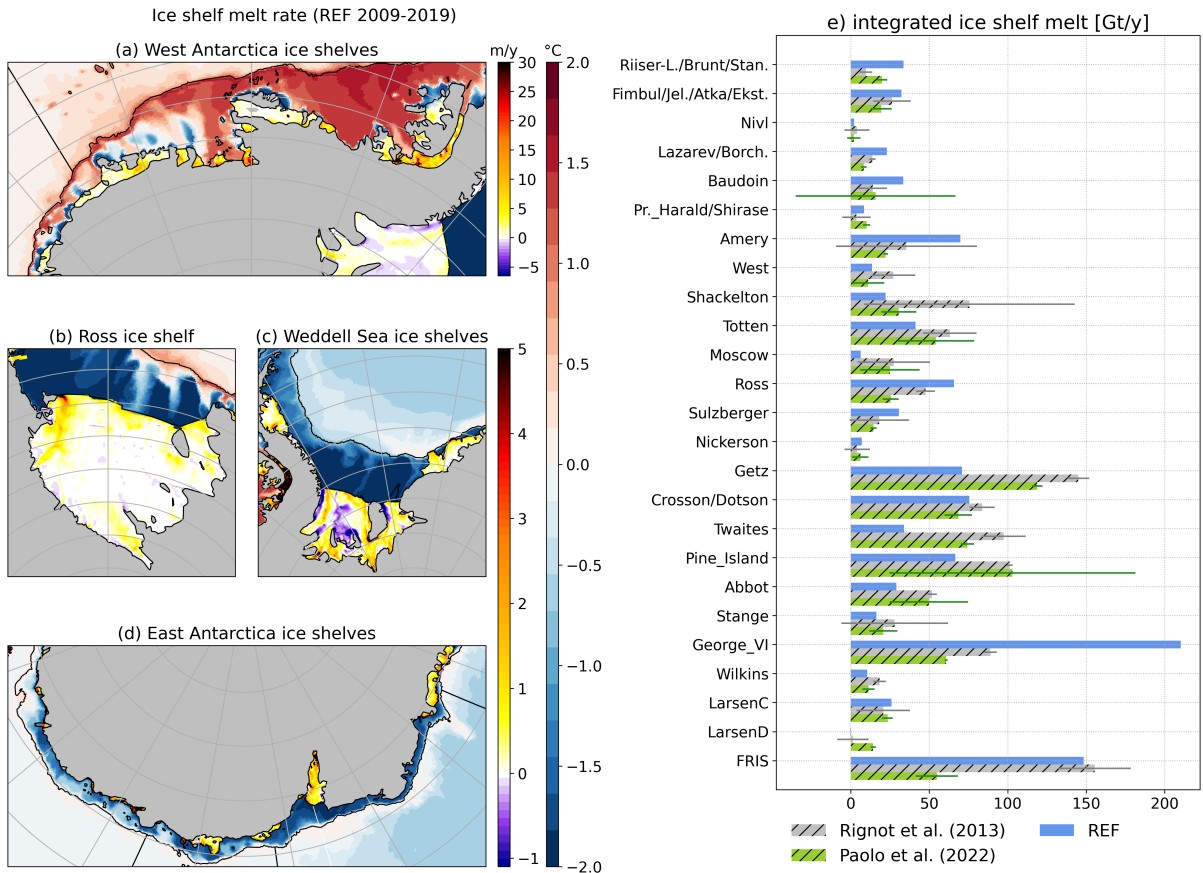

**Figure 5.** (a) to (d): Climatological ice shelf melt rate per sector (purple to red color map) with ocean bottom temperature on the Antarctic continental shelf (blue to red color map) in REF (2009-2018). (e) Basal mass loss of individual ice shelves in REF (in $\mathrm{Gt\,yr^{-1}}$) averaged over the period 2009-2018. The observation-based estimates from Rignot et al. (2013) and Paolo et al. (2023) are respectively in black and gray. For clarity, the ice shelves smaller than 4000 $\mathrm{km}^2$ in Rignot et al. (2013) are not represented. See Rignot et al. (2013)'s Fig. 1 for the locations of individual ice shelves.

the one visible in satellite data (Shean et al., 2019), although simulated melt rates there are underestimated by approximately a factor of two. This underestimation near the grounding line may be due to a lack of horizontal and vertical resolution in this area (the melt pattern is more realistic at $1/12°$, Jourdain et al., 2022) as well as the absence of subglacial runoff (Nakayama et al., 2021). Another noticeable bias is the absence of refreezing area beneath Amery Ice Shelf compared to satellite products (Wen et al., 2010; Rignot et al., 2013; Adusumilli et al., 2020), possibly related to the aforementioned lack of polynya activity upstream of Amery Ice Shelf.

### 3.4 Interannual variability in the Amundsen Sea

The vertical stratification as well as the interannual to decadal variability of ice shelf basal melt rates as well as ocean properties in front of the ice shelves are well documented for the Amundsen Sea thanks to recurrent oceanic cruises (e.g., Jacobs et al., 1996; Dutrieux et al., 2014; Jenkins et al., 2018). The simulated temperature profiles near Dotson and Pine Island are overall within the interannual observational range, although the simulated thermocline is sharper and shallower than observed (Fig. 6e,f). Our REF simulation captures the transition to a relatively warm period between approximately 2005 and 2010, although the prior and posterior cold states remain significantly warmer than observed (Fig. 6c,d). As a consequence, the interannual variability of ice shelf melting is underestimated for both Dotson and Pine Island (Fig. 6a,b).

## 4 Twenty-third century SSP5-8.5 perturbation

### 4.1 General circulation and sea ice

The ocean barotropic circulation undergoes a strong adjustment to the perturbation in the first five to ten years, followed by a slower drift to nearly a steady state reached after approximately 80 years of perturbation (Fig. 7). The Weddell Gyre strengthens by $\sim 30\%$ (Fig. 7a) and extends further east, reaching Prydz Bay. The westward slope current constituting the southern flank of the gyre is highly intensified across the Weddell Sea (Fig. 8). The Ross Gyre is doubled in intensity (Fig. 7b) and extends further east, reaching the Bellingshausen and Amundsen Seas (Fig. 8). This is consistent with changes in wind stress curl due to changes in the atmospheric circulation (Fig. 1e) and sea ice loss, as previously reported in projections over the $21^{st}$ century (Gómez-Valdivia et al., 2023). The ACC transport decreases to 110-115 Sv (Fig. 7c), likely due to a shutdown of HSSW and AABW formation as very little sea ice is produced in the perturbed experiment. There is indeed no sea ice left in summer and the maximum sea ice area declines from 18 million $km^2$ to less than 1 million $km^2$ in September (not shown).

### 4.2 Water mass properties on the continental shelf

The general picture is that sea ice production becomes insufficient to maintain HSSW on the continental shelf, which decreases the density barrier between cold shelf water and CDW, thereby enabling CDW flows onto the continental shelf (Naughten et al., 2021). This is particularly visible in the first 10 years of the perturbed simulations, at the Ronne depression (WWED box in Fig. 9b) and the Victoria Land Basin (WROSS box in Fig. 9d), two major sites of HSSW formation. CDW intrusions first

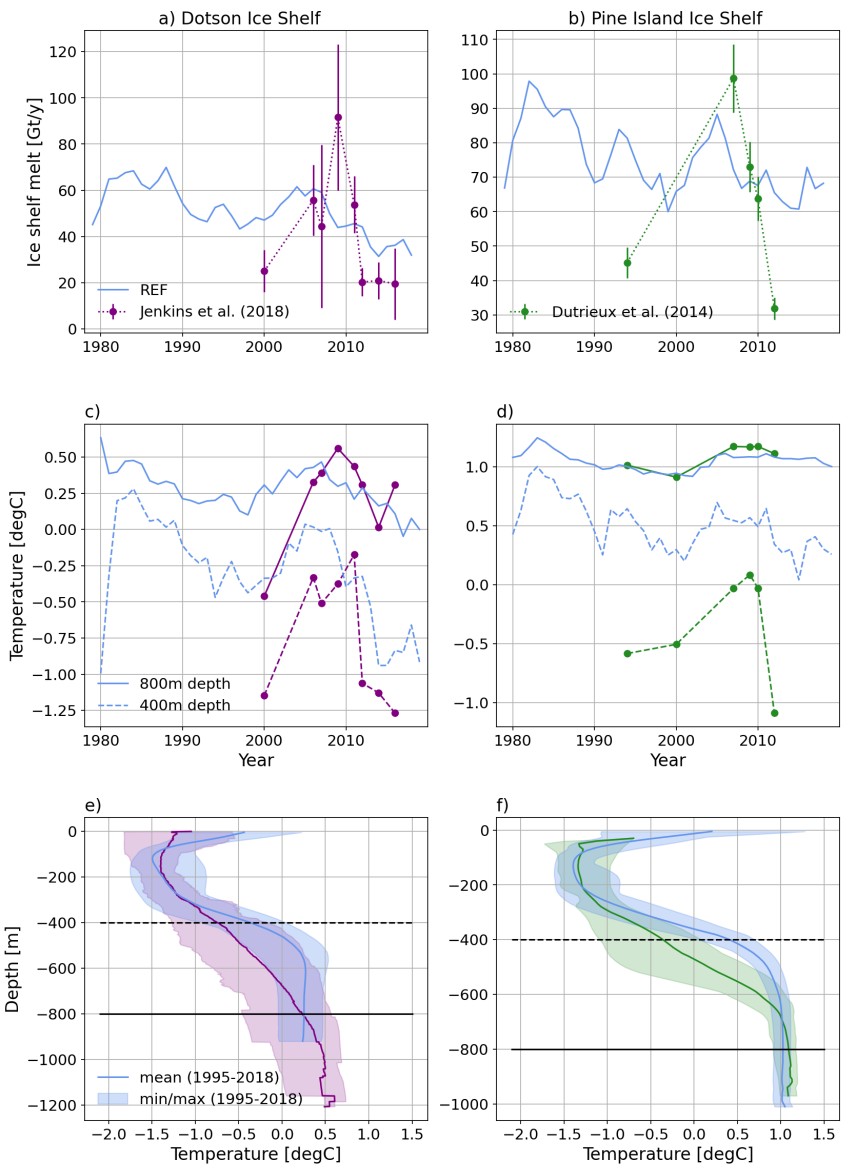

**Figure 6.** Model evaluation of Dotson (left column) and Pine Island (right column) ice shelves. Simulated properties are in blue while observational estimated are in red for Dotson (Jenkins et al., 2018) and in green for Pine Island (Dutrieux et al., 2014). (a,b) Melt timeseries, (c,d) potential temperature timeseries at 400 and 800 m depth in front of the ice shelves, and (e,f) mean December to February temperature profile between 1995 and 2018 and near the ice shelf front, as well as the minimum-maximum interannual range (shaded).

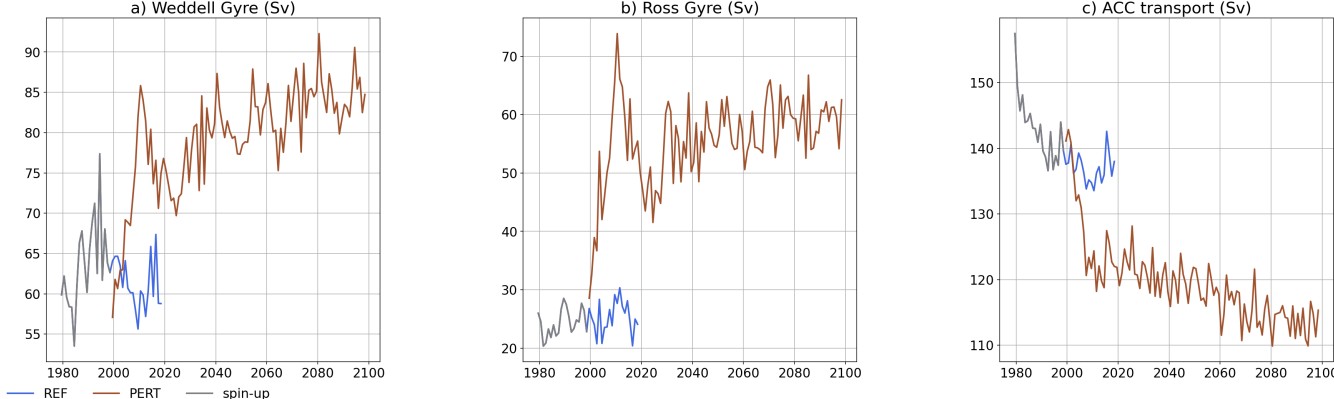

**Figure 7.** Time series of the Weddell Gyre (a), Ross Gyre (b), and ACC transport (c) in REF (brown) and PERT (blue).

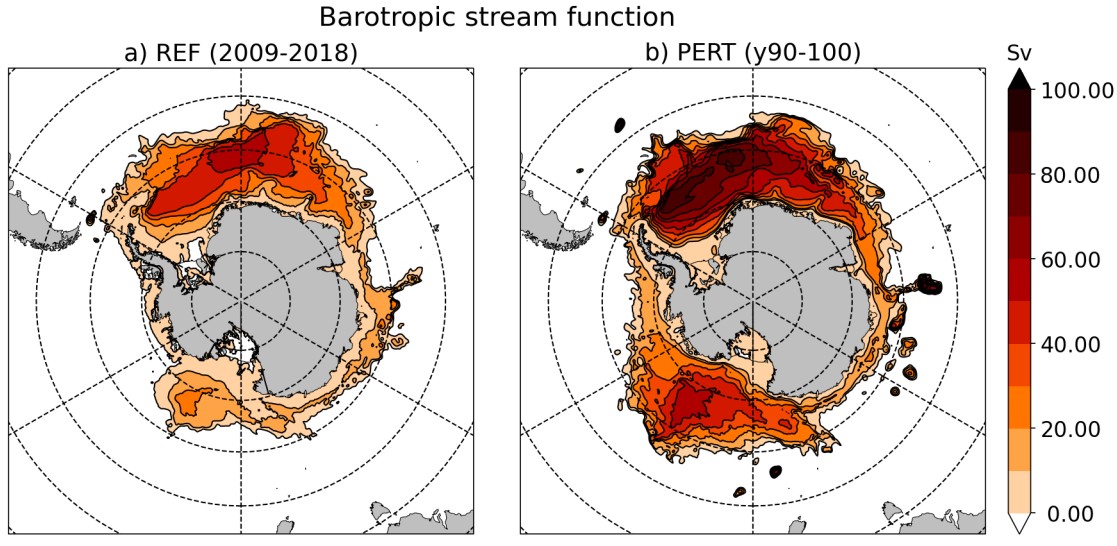

**Figure 8.** Climatological barotropic stream function in REF (a) and PERT (b). In white, area beyond the polar gyres. Each contour is 10 Sv.

occur on the eastern part of these seas, in the Filchner Trough (EWED box in Fig. 9a) and Little America Basin (EROSS box in Fig. 9c). The rapid initial adjustment to the new forcing is followed by a slow trend toward freshening and warming, which can be explained by slow changes in deep ocean properties at the circum-Antarctic scale (Sallée et al., 2013).

The Amundsen Sea becomes warmer than present-day conditions within 20 years, with very slow increase afterwards (Fig. 9e). By the end of perturbation, bottom temperatures warm by 2°C on the Amundsen Sea shelf. This is a much stronger
warming than those obtained by Caillet et al. (2022) from local atmospheric perturbations typical of 2300, likely due to the expansion of the Ross Gyre in our simulations (Fig. 8) while Caillet et al. (2022) used constant far-field ocean circulation. In a warmer climate, the Ross Gyre is indeed projected to grow towards the Amundsen-Bellingshausen Seas, which favours CDW intrusion onto the continental shelf, leading to a subsurface warming that may exceed 1°C by 2100 on the continental shelf (Gómez-Valdivia et al., 2023).

To further illustrate the processes in place, we now compare a section across the Dumont d'Urville Sea (north of Adélie Land) with a section in the Eastern Amundsen Sea, which can presently be classified as "dense shelf" and "warm shelf", respectively (Thompson et al., 2018). The former is characterised by cold and dense/salty water on the shelf and the second by the presence of weakly modified CDW at depth (Fig. 10). In the perturbed experiment, both locations are characterised by the presence of CDW (typical of a "warm shelf") and very strong vertical and northward density gradients both typical of a
"fresh shelf" (Thompson et al., 2018). We therefore suggest to classify this as "warm–fresh shelf", in which the cold Antarctic Surface Water usually found over fresh shelves is replaced by a fresh and warm water mass overlaying a saltier CDW layer. The very strong density gradients in the perturbed experiments are a result of decreased sea ice production and subsequent convection, combined with increased ice-shelf melting and precipitation, as well as icebergs melting closer to Antarctica.

In the perturbed scenario, the circumpolar zonal winds are shifted southward (Fig. 1e,f), which increases sea surface height
over the continental shelf through southward Ekman transport (Fig. 10e,f), as previously reported by Spence et al. (2014) in future projections. In the Amundsen and Bellingshausen seas, changes in the Ekman transport are also linked to more cyclonic wind in a warmer climate (Fig. 1 and Gómez-Valdivia et al., 2023). By geostrophy, this wind perturbation induces a westward surface current above the shelf break (Fig. 10). The strong northward density gradient due to the shelf freshening have an opposite effect that accumulates with depth ("thermal wind" effect), which tends to cancel or revert eastward the current near
the sea floor (Fig. 10). This weak or eastward zonal current near the sea floor at the shelf break favors the intrusion of CDW onto the shelf (e.g., Wåhlin et al., 2012; Walker et al., 2013).

Our results are quite different from Spence et al. (2014) who found a flattening of the isopycnals in response to poleward shifting winds. This is likely because they did not represent the changes in air temperature, precipitation and ice shelf melt. Our proposed mechanism is more similar to the baroclinic response proposed by Silvano et al. (2022), except that the increased
density gradient is not only due to the accumulation of surface water on the shelf through Ekman transport, but also to the coastal freshening induced by decreased sea ice production, increased ice-shelf melting and increased precipitation.

The only area of the continental shelf that remains cold after 100 years of perturbation is the Ronne depression (Fig. 11c), where a specific mechanism is at play. In the reference simulation, significant amounts of HSSW are formed in the Ronne depression and flow under the Ronne Ice Shelf (Fig. 12a). In the perturbed experiment, the Ronne depression is occupied by

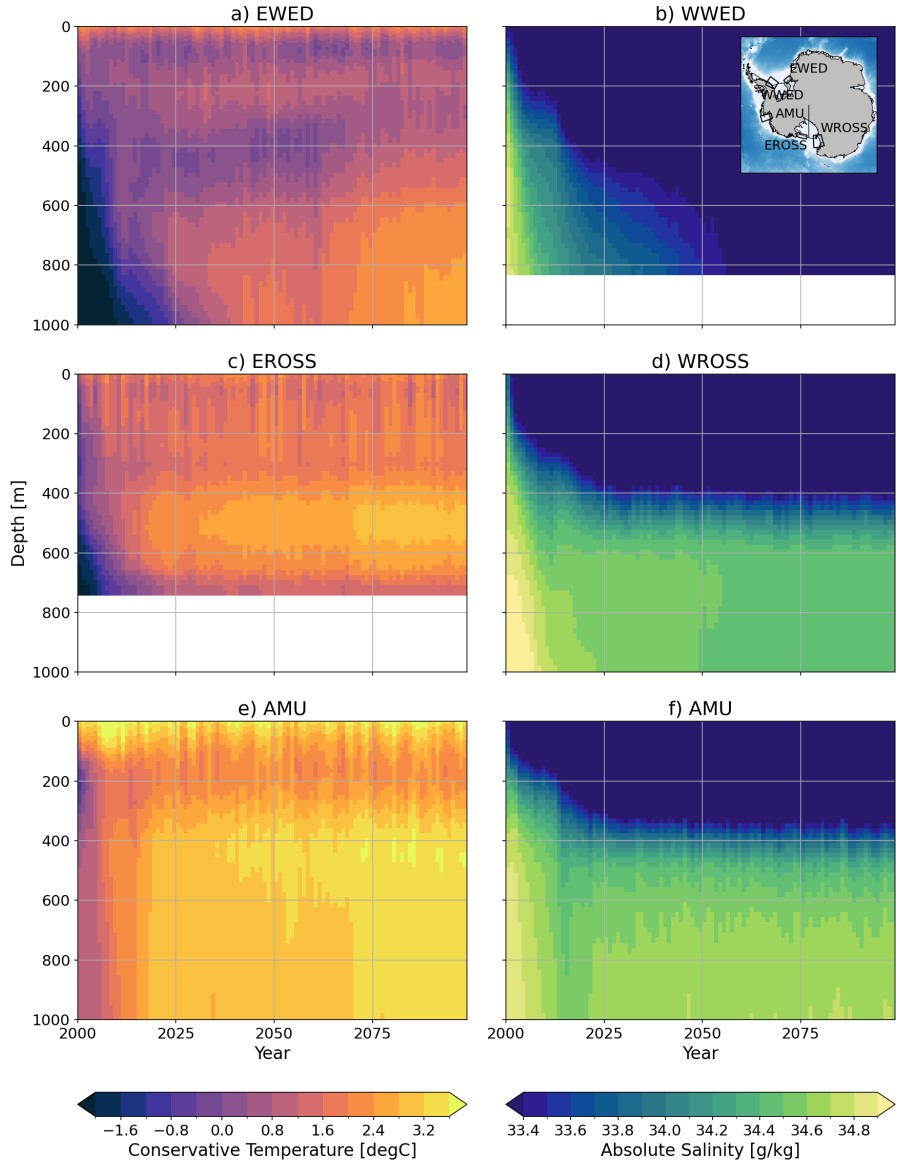

**Figure 9.** Left: temperature vertical profile as a function of time in PERT, averaged in East Weddell box (a), East Ross box (c) and Amundsen sea box (e). Right: salinity vertical profile as a function of time in PERT, averaged in West Weddell box (a), West Ross box (b) and Amundsen sea box (c). See inset in panel (b) for box definitions.

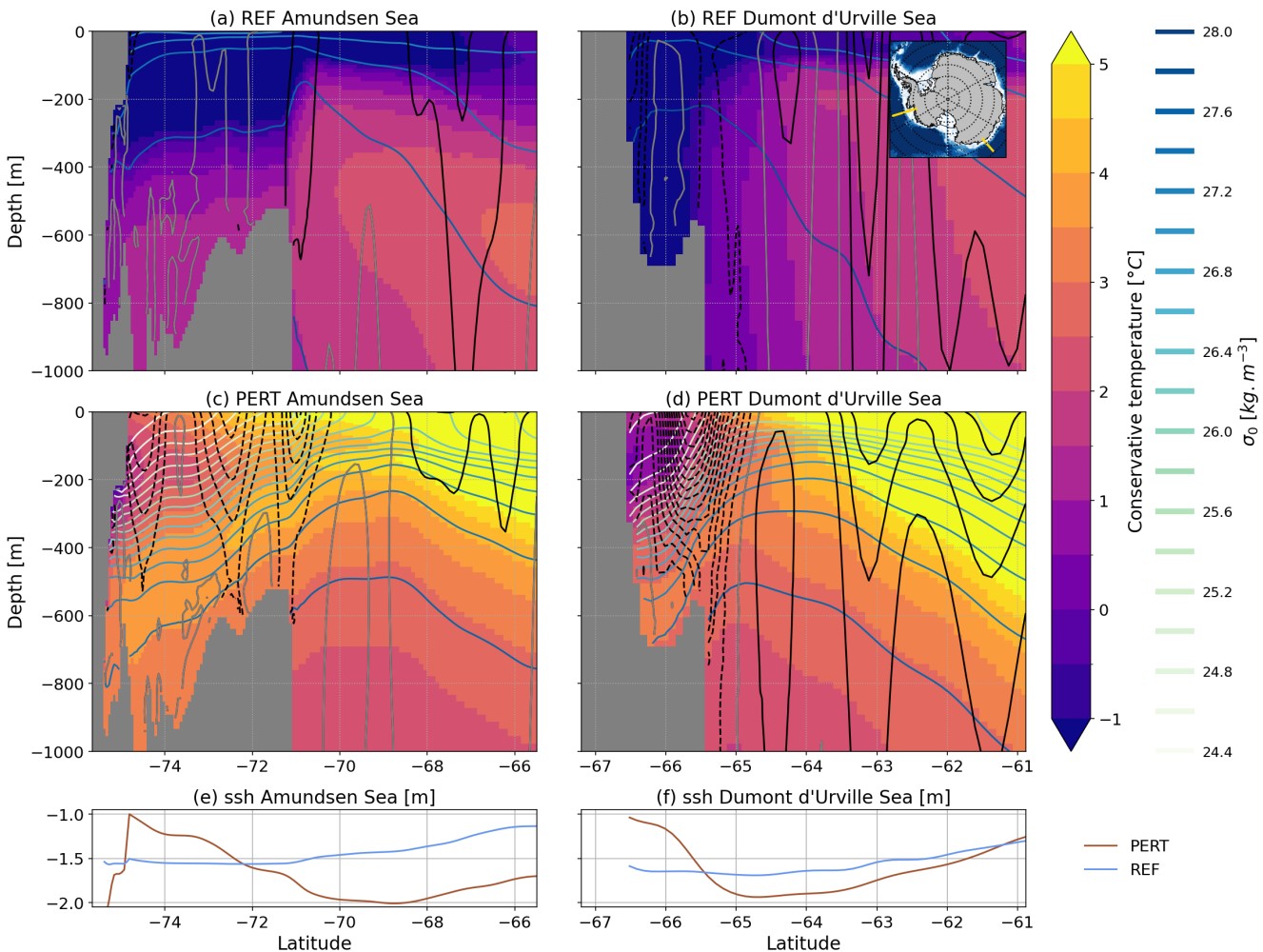

**Figure 10.** (a)-(c) Temperature section across the Amundsen Sea. (b)-(d) temperature section across the Dumont d'Urville Sea. The colored contours are isopycnals and the black contours show the zonal velocity every 5 cm s$^{-1}$ (dashed/solid lines for westward/eastward velocity, gray line for zero velocity). (e)-(f) Sea surface height profile along the corresponding sections (blue for REF and brown for PERT).

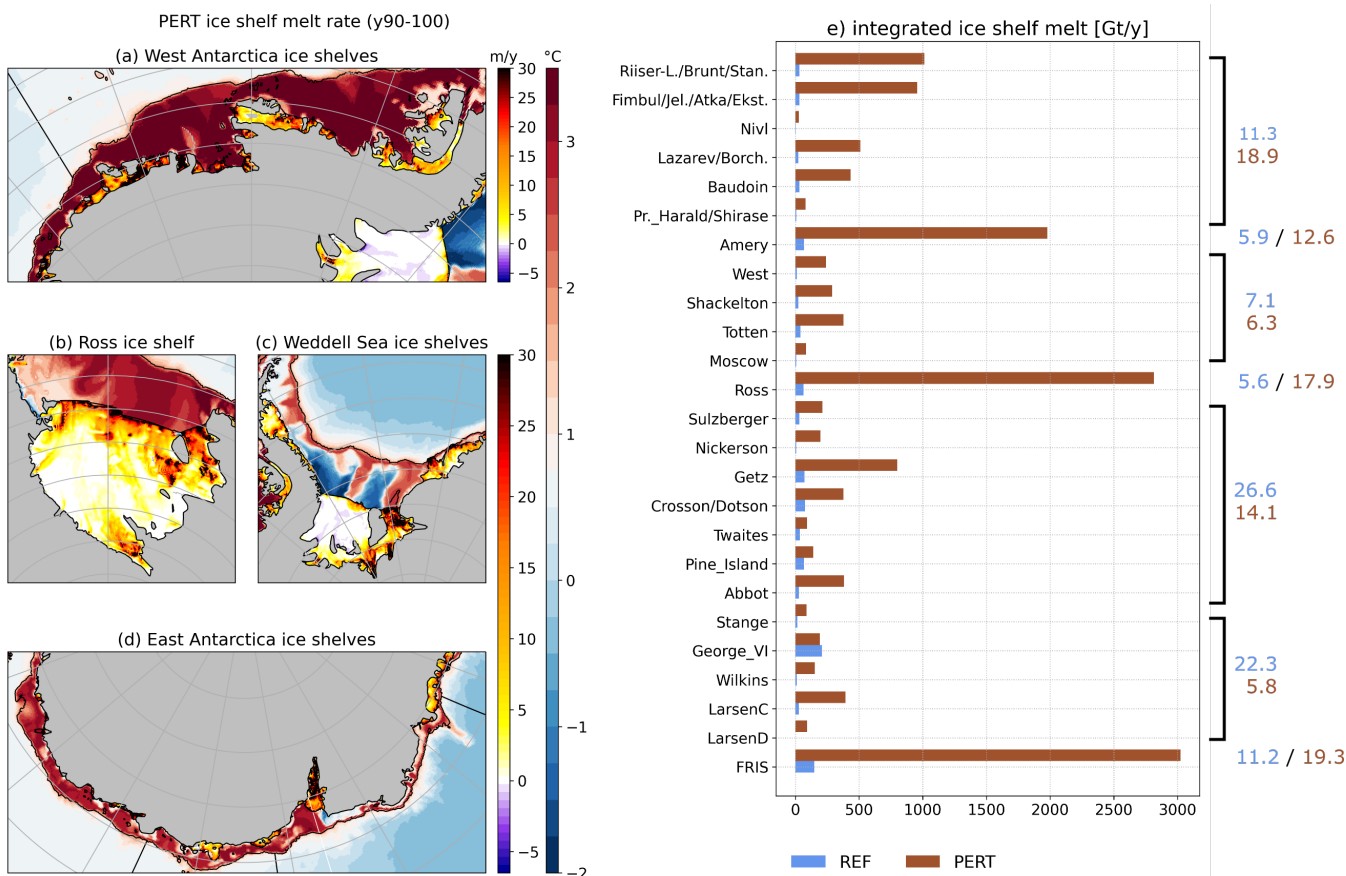

**Figure 11.** (a) to (d): Climatological ice shelf melt rate per sector (purple to red colormap) with bottom temperature on the Antarctic continental shelf (blue to red colormap) in PERT after 100 years of perturbation. (e) Total basal melt per ice shelf in Gt yr$^{-1}$ in PERT (blue) and REF (brown). For clarity, only ice shelf larger the 4000 km$^2$ in Rignot et al. (2013) are represented. Numbers in (e) indicate the relative contribution of individual ice shelves with respect to the total Antarctic melt in REF (blue) and PERT (brown).

water flowing out of the Ronne cavity and coming all the way from the Filchner trough and Central Trough (Fig. 12b). Despite a strong inflow of warm water into the Filchner and central troughs, there is still refreezing underneath Ronne (see following subsection) and water colder than surface freezing point is produced and exported out of the cavity (Fig. 12c). This indicates that all the heat that comes into the Filchner-Ronne cavity is consumed to melt the ice shelf even in a much warmer climate. It should be noted that this presence of cold outflow was not found by Naughten et al. (2021) in their abrupt-4xCO2 experiments corresponding to a lower warming level than in our study.

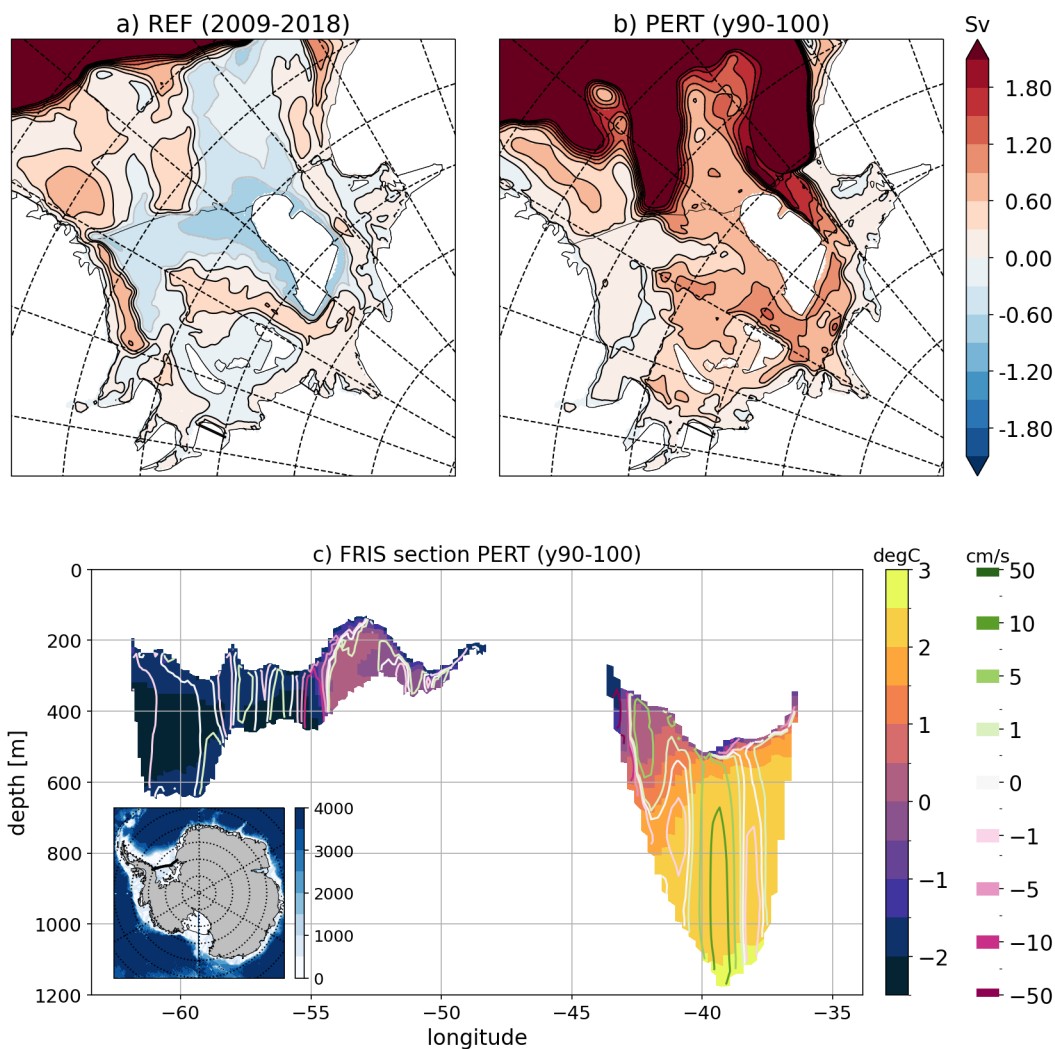

**Figure 12.** Upper: Barotropic stream function (in Sv, see caption of Fig. 2) under Filchner Ronne Ice Shelf in REF (a) and PERT (b). Lower: (c) Ocean temperature section along Filchner Ronne ice shelf front in PERT (see thick black line on the map inset). Contours are the velocities normal to the section (positive toward the cavity and negative toward the open ocean).

## 4.3 Ice shelf melt rates

As expected, ice shelf basal melt rates follow the same trend as bottom temperatures on the nearby continental shelf in response to the forcing perturbation. The total Antarctic melt increases 11 folds, from 1,180 to 15,700 $\text{Gt yr}^{-1}$. The Antarctica averaged melt rate increased from 0.80 $\text{m yr}^{-1}$ to 10.64 $\text{m yr}^{-1}$ (meter water equivalent).

All present-day cold cavities, such as Ross, Amery and Filchner, become warm in the perturbed experiment, and melt rates reach levels similar to those currently observed in the Amundsen Sea (Fig. 11). The present-day warm cavities, in the Amundsen and Bellingshausen Seas, also experience increased melt rates, which are explained both by the warming resulting from the eastward extension of the Ross Gyre (Fig. 8 and Gómez-Valdivia et al., 2023), and by the strong warming of the winter mixed layer (between 100 and 250 m depth in Fig. 9c) resulting from the strong reduction of sea ice production in winter. The melt increase is particularly strong for Abbot and Getz ice shelves (Fig. 11) because a large portion of the ice draft is currently located in the cold winter mixed layer (Cochran et al., 2014; Wei et al., 2020) and experiences a shift to much warmer conditions in the perturbed experiment. Given that the position of the thermocline with respect to the ice draft largely drives the transition to a high melting regime, we believe that the ice draft correction applied to Getz has made the transition more realistic.

The Ross and Pine Island ice shelves experience a sharp increase in basal melt rates in the first years of perturbation, followed by a stabilization. As the Ross continental shelf, changes from cold (near -2°C) to warm (> 2°C) conditions, the relative change in thermal driving is very large and melt rates are multiplied by ∼30, reaching 2,810 $\text{Gt yr}^{-1}$ (Fig. 13). This is three times more than the 900 $\text{Gt yr}^{-1}$ obtained by Siahaan et al. (2021) at the end of the 21[st] century under SSP5-8.5 with a Ross continental shelf at ∼2°C. Despite this strong warming, the Ross ice shelf still exhibits a stable amount of refreezing (1 $\text{Gt yr}^{-1}$) after the first decade (not shown).

As the Amundsen Sea is already warm in present-day conditions, the thermal driving at depth only doubles so that Pine Island and Thwaites experience weaker relative increase in melt rates. For Crosson, Dotson and Getz, the present-day ice shelf is partly located in the thermocline, so that a part of these ice shelves experience larger relative increase in thermal driving. This results in an additional basal mass loss of ∼1,000 $\text{Gt yr}^{-1}$ due to the perturbation, which is much higher than the additional 350 $\text{Gt yr}^{-1}$ obtained by Jourdain et al. (2022) for the entire Amundsen sector at the end of the 21[st] century under SSP5-8.5.

The Filchner-Ronne Ice Shelf exhibits a distinct behaviour: melt rates increase steadily during 100 years, but there is still a stable and significant amount of refreezing (26 $\text{Gt yr}^{-1}$) in the last decades of our experiment (Fig. 11). After 100 years, the basal mass loss of Filchner-Ronne reaches 3,000 $\text{Gt yr}^{-1}$, i.e., ∼20 times larger than in REF. In comparison, Naughten et al. (2021) simulated a peak mass loss of 1,600 $\text{Gt yr}^{-1}$ for similar bottom temperatures in the Filchner trough (∼2.4°C), while Haid et al. (2022) obtained a peak at 1,800 $\text{Gt yr}^{-1}$ for ∼1.7°C.

Two aspects may explain these different sensitivities. First, we simulate strong increase in melt rates near the ice shelf front because of the disappearance of the cold surface layer in our simulations. In the two other studies, the surface layer is still cold and the presence of an interactive ice sheet allows the ice shelf to thin and thereby to partly remain in this cold layer. Second, our parameterisation of tide-induced mixing in the three-equation system (Jourdain et al., 2019) may have a significant effect

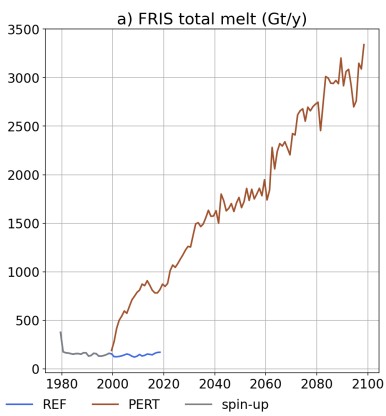 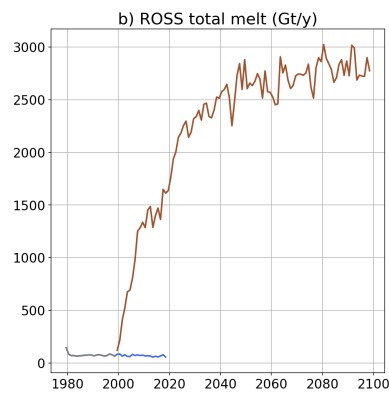 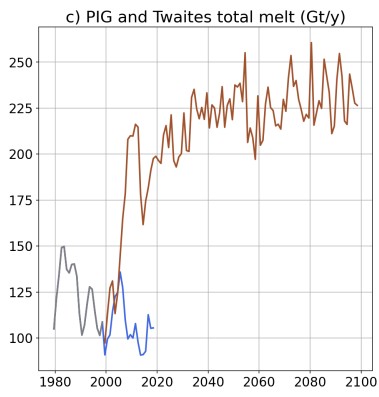

REF —— PERT —— spin-up

**Figure 13.** Total ice shelf melt time series in PERT for Filchner-Ronne Ice Shelf (a), Ross Ice Shelf (b) and Pine Island and Thwaites ice shelves (c).

on the Ross and Ronne-Filchner melt rates. While this parameterisation has a weak effect in present-day conditions because all the available heat is consumed anyway (Hutchinson et al., 2023), the abundance of warm water in the future may enhance its role in the largest ice shelf cavities.

As all the shelves turned warm, the main contributors to the total ice shelf melt in PERT are mainly the one with the largest area. In REF, the West Antarctic Ice Sheet and Peninsula sectors are responsible for 49% of the total melt for only 15% of 340 the total Antarctic ice shelf area. In PERT, this contribution falls to 20%. In contrast, the three giant cold ice shelves (Ross, Filchner-Ronne and Amery) are responsible for 50% of the total melt in PERT versus 23% in REF, for 65% of the total area (Fig. 11).

## 5 Conclusions

In this study, we have presented an new set-up for the global NEMO configuration at 0.25° resolution (eORCA025). Thanks 345 to a preliminary tuning of the sea ice model parameters and of the lateral and bottom boundary conditions at the northern end of the Antarctic Peninsula, we simulate realistic water masses in the Southern Ocean and on the Antarctic continental shelf. This is important as the performance of previous versions of eORCA025 was not good enough to be used in ocean–ice-sheet simulations (Smith et al., 2021). The simulated basal mass loss of Antarctic ice shelves is $1,180\,\mathrm{Gt\,yr^{-1}}$ on average, which aligns well with the observational estimates. Simulating the interannual variability in the Amundsen Sea nonetheless remains 350 challenging, with an underestimated variability in our simulations.

We have then used this configuration to investigate the ocean and sea ice response to a strong and abrupt perturbation of the atmospheric conditions. To our knowledge, our study is the first to investigate plausible conditions for the late 23$^{\mathrm{rd}}$ century at the scale of Antarctica under a high-end scenario (SSP5-8.5 and high equilibrium climate sensitivity of the driving climate model). Our simulations reveal that the entire Antarctic continental shelf is subject to substantial warming within the first two

decades of perturbation, and several decades of adjustment for the largest ice shelf cavities. In particular, the Ronne-Filchner ice shelf has a response time exceeding 100 years.

Our perturbation experiment is idealised in many ways. The abrupt transition to late 23$^{\mathrm{rd}}$ century conditions does not account for slow changes in the global thermohaline circulation, including the formation of CDW very far from the Southern Ocean. At these timescales, interactions with the evolving ice sheet and atmosphere should also be taken into account for more realistic simulations (e.g., Donat-Magnin et al., 2017; Bronselaer et al., 2018; Bell et al., 2018). In this sense, future model intercomparison with more realistic perturbations under strong emission scenarios will be needed. We nonetheless believe that we have identified key mechanisms that set the primary characteristics of the ocean and ice shelf response to strong climate perturbations, as described hereafter.

Under warmer atmospheric conditions, the sea ice cover drastically diminishes, even during winter, and the production of HSSW ceases, resulting in a rapid freshening of the previously salty continental shelves. In the absence of HSSW production, the intrusion of CDW onto the former cold continental shelves (Ross, Weddell, and East Antarctica) becomes more pronounced, although the rate of change may vary in different locations. These alterations in oceanic properties lead to a substantial increase in ice shelf melt rates, with a total basal mass loss escalating from 1,180 Gt yr$^{-1}$ to 15,700 Gt yr$^{-1}$ after 100 years of perturbation. This significant increase is primarily attributed to the former colder ice shelves, which experience a substantial enhancement in thermal driving as they transition from cold cavities to warm cavities. In contrast, the relative change in ice melt rate for the warm cavities is comparatively smaller than that of the cold cavities.

*Code availability.* Source code and parameter namelists for each experiments, as well as script used to make the figures are all available here: https://github.com/pmathiot/paper_MJ2023 or here: https://doi.org/10.5281/zenodo.8411091.

*Data availability.* Data used to run the simulations are available upon request to the lead author. Monthly atmospheric forcing anomalies as well as monthly climatologies of the last 30y of PERT and REF are available on zenodo in Mathiot and Jourdain (2023)

*Author contributions.* PM conducted the numerical experiments, analysed and visualised the results and led the writing process; NJ contributed to the experimental design, the results analysis and to the writing process

*Competing interests.* The authors declare that no competing interest are present.

*Acknowledgements.* This study was funded by the European Union's Horizon 2020 research and innovation programme under grant agreement No 820575 (TiPACCs) and by by the French National Research Agency under Grant ANR-19-CE01-0015 (EIS). N. Jourdain was also

funded by EU-H2020 grants No 101003536 (ESM2025) and No 869304 (PROTECT). This work was granted access to the HPC resources of CINES and TGCC under allocations A0100106035 and A0120106035 attributed by GENCI.

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
