# Peer review of "Southern Ocean warming and Antarctic ice shelf melting in conditions plausible by late 23rd century in a high-end scenario"

_EGUsphere, 2023_

## Author Comment (AC1)

In blue: reviewer comments
In black: authors reply

**General comments**

I very much enjoyed reading this study and my suggested revisions are all minor. It has a nice balance between building on previous work (eg expansion of the Ross Gyre, tipping of the FRIS cavity into a warm state) and exploring uncharted territory by warming the entire continent outside the bounds of what has been tested before. The new configuration of NEMO is also a major advance, and in places the tuning choices need more explanation (see my specific comments below). The processes responsible for warming and ice shelf melting in each sector are only explored briefly, but this is probably appropriate given the circumpolar approach and the references to previous work.

*Authors reply:* We thank Kaitlin Naughten for her positive comments. These comments and those from Reviewer #2 have prompted us to go further into the description of the mechanisms. We have also tried to better explain the tuning choices, and overall, we believe that our manuscript has greatly improved through this review process.

I hope that future work will build on these simulations by analysing the sector changes in more detail and using the results to drive ice sheet models.

*Authors reply:* As part of the TiPACCs project that funded a part of this work, the next stage for us is to analyze the effect in a coupled ocean/ice sheet model.

I feel the paper could do more to position the simulation as an idealised change or hypothesis test, rather than an outcome which is plausible for the future. Between the fossil fuel scenario, the time frame, and the high sensitivity climate model used for forcing, this is an extreme upper bound for what we might expect in the real world. The uncoupled atmosphere and ice sheet also introduce substantial uncertainty, as well as the step-change nature of the forcing. This simulation is still very useful for our theoretical understanding of Antarctica, but I would hesitate to consider it a "projection".

*Authors reply:* We believe that this is a projection in the sense that it starts from present conditions and explore the response to high-end atmospheric perturbations related to high-end anthropogenic emissions. We also consider that the absence of coupled atmosphere or ice-sheet should not prevent us from calling it a projection given that the "CMIP projections" neither represent ice shelf cavities nor include an interactive ice sheet, and the "ISMIP projections" have neither ocean nor atmosphere coupling. We nonetheless agree that we are much closer to the storyline concept introduced by Shepherd et al. (2018) than to a simulation that can be used in an ensemble to estimate probabilities. We think that "*high-end*" in the title and throughout the text makes it quite clear that we are really at the upper boundary of something plausible, but according to IPCC-AR6, neither SSP5-8.5 nor an ECS of 4.6°C can be ruled out even if their probability is very low. Given the step change used in our simulation and other limitations, we have nonetheless replaced "projections" with "idealized projections", or we mention plausible conditions in a high-end, extremely unlikely, late 23[rd] century, throughout the revised manuscript.

Shepherd, T. G., Boyd, E., Calel, R. A., Chapman, S. C., Dessai, S., and others (2018). Storylines: an alternative approach to representing uncertainty in physical aspects of climate change. *Climatic change*, 151, 555-571.

There is very little discussion of the Amery Ice Shelf, but from the figures it appears to experience the same mechanism of tipping as the Ross and FRIS. If this is the case, it is the

first simulation of Amery tipping to my knowledge, and this warrants more attention in the text.

*Authors reply:* yes, it is not specifically mentioned that Amery tipped because all the Antarctic shelves are becoming warm or warmer. We nonetheless start the 2nd paragraph of section 4.3 as: *'All present-day cold cavities, such as Ross, Amery and Filchner, become warm in the perturbed experiment, and melt rates reach levels similar to those currently observed in the Amundsen Sea'.*

The paper needs more discussion of glaciological implications, perhaps at the very end. Which marine basins of Antarctica would be threatened by these changes (all of them?), and what combined sea level equivalent could be at risk from basal melting? Do we have any idea of the timescale of response? Of course the details cannot be answered by the current study, but some exploration of the implications would be welcome. A brief discussion of potential feedbacks between ice sheet geometry and the ocean state would also be suitable here, as a very retreated ice sheet would surely change the total melt flux.

*Authors reply:* We mention the importance of the ice sheet feedback in the conclusion. It is very difficult to compare our melt rates to other ice sheet modelling studies, first because very few provide their basal melt rates in 2300, then because the diminution of the ice shelf area and thickness in ice sheet models alter the integrated values. For example, the 15,700 Gt yr$^{-1}$ reached in our perturbed experiment in conditions of 2300 correspond to the upper end (95th percentile) of the ice shelf basal mass loss in 2200 in Coulon et al. (2023, under review). But their 2200 estimate is for altered ice shelves, and it decreases after 2200 due to the increasing number of collapsed ice shelves. For this reason, we prefer not to speculate on the details of the ice sheet response and its feedback to the ocean.

Coulon, V., Klose, A. K., Kittel, C., Edwards, T., Turner, F., Winkelmann, R. and Pattyn, F. (2023). Disentangling the drivers of future Antarctic ice loss with a historically-calibrated ice-sheet model. *EGUsphere*, 2023, 1-42.

**Specific comments**
Title: change "typical of" to "possible by". How can we say what is "typical" of a time period that hasn't happened yet?

*Authors reply:* We changed the title to 'Southern Ocean warming and Antarctic ice shelf melting in conditions plausible by late 23rd century in a high-end scenario'

Line 4 (abstract): change "typical of" to "projected by", for the same reasons as above.
*Authors reply:* We decided to use again '*plausible by*' as in the title.

Lines 16-23: The first paragraph of the introduction needs a bit more fleshing out. How do ice sheet models infer basal melting from climate simulations (I understand there's a few different approaches, eg nearest neighbour SST or averaging over the continental shelf), and why are these the wrong processes? The casual reader would probably not follow this as written.

*Authors reply:* Our first paragraph has been rewritten as follows:

'*Most future projections of the Antarctic contribution to sea level rise have so far relied on ice sheet models in which ice shelf basal melt was parametrised from the changing ocean characteristics of global climate simulations (e.g., Cornford et al., 2015; Seroussi et al., 2020; Levermann et al., 2020; DeConto et al., 2021; Payne et al., 2021). Such parametrisations calculate ice shelf basal melt rates from the ocean properties on the continental shelf and do not explicitly represent the ocean circulation and mixing in ice shelf cavities, including the crucial interactions with bathymetric features and*

*tides (Burgard et al., 2022). They are directly fed by the outputs of global climate simulations that are highly biased near Antarctica (Little and Urban, 2016; Barthel et al., 2020), partly due to their coarse resolution (van Westen and Dijkstra, 2021) and to the absence of feedbacks between glacial meltwater and the climate system (Donat-Magnin et al., 2017; Bronselaer et al., 2018; Sadai et al., 2020; Li et al., 2023). For these reasons, a number of modelling centers are currently incorporating interactive Antarctic Ice Sheet models into their climate models (e.g., Smith et al., 2021; Pelletier et al., 2022). For this, the ocean components of climate models need to represent the ocean circulation beneath ice shelves.'*

Line 43: Can you summarise in 3 words what this bug related to? The current text sounds a bit alarming, and not all readers will go and track down the ticket.

*Authors reply:* The bug consisted of a typo that changed drastically the distribution of solar and non-solar fluxes over ice covered areas. As a consequence, the ice cover and volume increased and some processes such as lateral sea-ice melting had wrong seasonality. The text has been modified as follows:

> *'The ocean model used in this study is based on NEMO version 4.0.4, which represents the ocean dynamics and physics (NEMO-OPA, NEMO System Team, 2019) and the sea ice dynamics and thermodynamics (SI3, NEMO Sea Ice Working Group, 2019). The migration from 4.0.3 to 4.0.4 version contained a critical bug on the distribution of solar and non-solar fluxes over sea-ice covered areas but this was fixed in the version used in this study (complete description of the bug available on https://forge.ipsl.jussieu.fr/nemo/ticket/2626).'*

Lines 52-54: Thinning the Getz is an unusual way to compensate for a high melt bias. Is the Getz draft poorly constrained by data, which could somewhat justify this choice?

*Authors reply:* We have added more explanations in section 2.1:

> *'After preliminary tests, the Getz ice shelf draft was artificially thinned by 200 m (keeping the grounding line unchanged) in order to compensate a longstanding bias in the thermocline depth (previously reported by Mathiot et al., 2017). The later was driving very excessive release of meltwater, which was strongly deteriorating the mean state of the Ross Sea (a connection previously described in Nakayama et al., 2020). More details on the impact of such correction are provided in Section 3.3.'*

Then in section 3.3:

> *'The total melt underneath Getz was strongly overestimated in preliminary simulations, reaching 400-500 Gt yr$^{-1}$ (not shown). By reducing the ice shelf draft of Getz (section 2), we have artificially displaced it into the model cold mixed layer, which gives more realistic melt rates. This empirical correction of the ice shelf draft is nonetheless slightly too strong because it was done prior to the completion of parameter tuning.'*

Then in section 4.3:

> *'The melt increase is particularly strong for Abbot and Getz ice shelves (Fig. 11) because a large portion of the ice draft is currently located in the cold winter mixed layer (Cochran et al., 2014; Wei et al., 2020) and experiences a shift to much warmer conditions in the perturbed experiment. Given that the position of the thermocline with respect to the ice draft largely drives the transition to a high melting regime, we believe that the ice draft correction applied to Getz has made the transition more realistic.'*

The shape of Getz ice shelf is not perfectly known, with uncertainties on the ice draft of approximately 100 m in BedMachine-Antarctica, but a part of the issue is also clearly related to a bias in our thermocline depth.

Lines 60-67: What is the physical justification or reason for changing the slip condition and bottom friction around the Antarctic Peninsula?

*Authors reply:* The text has been slightly changed to address this comment:
- Changes in Section 2: *'A free-slip lateral boundary condition on momentum is applied with no slip condition applied locally at Bering Strait, in the whole Mediterranean sea, along the West Greenland coast and around the south Shetland, Elephant and south Orkney islands (at the Northern end of the Antarctic Peninsula). This technique is a crude method to take into account the locally complex sub-grid scale bathymetry, and it affects water mass properties as explained in section 3.2.'*
- Changes in section 3.2: *'The north end of the Antarctic Peninsula also exhibits a cold bias in REF. Preliminary analyses during the tuning processes suggested that this bias was sensitive to the HSSW properties (worse when HSSW was not dense enough), to the treatment of the bathymetry, and to the lateral slip condition and bottom friction at the tip of Peninsula.'*

Line 95: Add "currently" before "negligible" as surface runoff will surely not be negligible in the extreme scenarios considered later.

*Authors reply:* 'currently' has been added. The limitation in the extreme scenarios is mentioned in the revised description of the perturbation: *'and that runoff from ice melting at the surface will remain zero. All these assumptions are unrealistic even for projections to 2100 (Seroussi et al., 2020; Kittel et al., 2021)'.*

Lines 97-98: The freshwater flux correction needs a bit more explanation and justification for readers unfamiliar with the model configuration. Why was this necessary?

*Authors reply:* We thank the two reviewers for pointing this out and we have added more details in the revised manuscript:

> *'On top of other freshwater fluxes (precipitation, runoff ...), a common practice in forced ocean models is to use some form of sea surface salinity restoring. This restoring is required because of the missing atmospheric feedbacks on humidity in forced models (for more details see Griffies et al., 2016). To make the model sensitivity analysis more robust, this corrective term was diagnosed from sea surface salinity restoring towards WOA2018 over the period 1999-2018 in a former simulation (the "REALISTIC" simulation described in Burgard et al., 2022) and applied as an additional climatological monthly freshwater flux in all our simulations.'*

Line 107: How is the SSP5-8.5 scenario extended beyond 2100? I expect it has a sustained level of very high fossil fuel emissions - is this even possible given available fossil fuel reserves?

*Authors reply:* This is beyond our expertise, and we refer to the estimation of Meinshausen et al. (2020). In summary for the main greenhouse gases: fossil fuel emissions (methane and carbon dioxide) peak in 2090. After 2100, the fossil fuel emissions are ramped down to zero until 2250. Land use methane emissions are kept constant after 2100 and ramp down to zero until 2150 for carbon dioxide.

Line 110: Presumably there is a trend in simulated global climate over 1979-2018. How does repeating this period influence the simulation?

*Authors reply:* Repeating the period is common practice in ocean modelling to force ocean model for a long period of time as in the OMIP protocol (Griffies et al., 2016). The 'jump' every 40 years does not impact the overall response as shown in the time series of figure 7, 9

and 13. The trends over 1979-2018 are weak compared to the perturbation. Simmons et al. (2017) show that there is almost no trend in surface air temperature southward of 60°S, while our perturbation is greater than 5°C and exceeds 20°C for some months and locations. Therefore, we are confident that the diagnosed signal was generated by the perturbation.

Simmons, A.J., Berrisford, P., Dee, D.P., Hersbach, H., Hirahara, S. and Thépaut, J.-.-N. (2017), A reassessment of temperature variations and trends from global reanalyses and monthly surface climatological datasets. Q.J.R. Meteorol. Soc., 143: 101-119. https://doi.org/10.1002/qj.2949

Figure 1: I struggled to interpret the zonal wind changes visualised in panel e), especially the negative values on the continent. Perhaps anomaly vectors, and/or plotting the reference state, would help.
*Authors reply:* We have changed the panel with a vector plot of wind speed anomaly on top of the wind curl. It is much clearer. Furthermore, we figured out that the non-centered colormap was misleading in the negative value. It is no more the case with the vector plot.

Line 163: Change "requires" to "would require" to make it clear that this iceberg and fast ice physics does not exist in this version of NEMO.
*Authors reply:* DONE

Figure 4: Adding a third column of anomaly panels would make it easier to identify the model biases in temperature and salinity.
*Authors reply:* DONE

Lines 207-213: This short section should be expanded, to explore the possible reasons for underestimated variability. Does your bathymetry consider grounded icebergs on Bear Ridge (which Bett et al. 2020, doi:10.1029/2020JC016305 found was crucial to simulate colder conditions in the western Amundsen Sea)? Perhaps the polynya activity is insufficient, or the mixed layer salinity is biased low?
*Authors reply:* We thank the reviewer for highlighting this point. The line of icebergs grounded on Bear Ridge is actually present in our configuration. The revised model description now includes this information:
> '*Because of its effect on sea ice and water masses (mean state and variability) in the Amundsen Sea (Bett et al., 2020), the line of icebergs grounded on Bear Ridge has been added as land points blocking the advection of sea ice.*'.

Polynya activity seems also insufficient in general (not only in Amundsen Sea). It is our explanation on the why we had to decrease the maximum ice fraction to 0.95. By decreasing the maximum ice fraction, we artificially increase the air-sea interaction in the pack ice to correct the missing or not active enough polynya.

Lines 233-235: Summarise why an expanded Ross Gyre leads to a much warmer Amundsen Sea than local changes in onshore transport and modification, for those readers who are not familiar with the Gomez-Valdivia study.
*Authors reply:* Details have been added to the text:
> '*The Ross Gyre is doubled in intensity (Fig. 7b) and extends further east, reaching the Bellingshausen and Amundsen Seas (Fig. 8). This is consistent with changes in wind stress curl due to changes in the atmospheric circulation (Fig. 1e) and sea ice loss, as previously reported in projections over the 21st century (Gómez-Valdivia et al., 2023).*'

Lines 266-268: Siahaan et al. had a much coarser resolution, which could explain their weaker response of Ross melt rates.

*Authors reply:* We have added a new paragraph discussing the different sensitivity compared to Siahaan et al. and Haid et al.:

> '*Two aspects may explain these different sensitivities. First, we simulate strong increase in melt rates near the ice shelf front because of the disappearance of the cold surface layer in our simulations. In the two other studies, the surface layer is still cold and the presence of an interactive ice sheet allows the ice shelf to thin and thereby to partly remain in this cold layer. Second, our parameterisation of tide-induced mixing in the three-equation system (Jourdain et al., 2019) may have a significant effect on the Ross and Ronne-Filchner melt rates. While this parameterisation has a weak effect in present-day conditions because all the available heat is consumed anyway (Hutchinson et al., 2023), the abundance of warm water in the future may enhance its role in the largest ice shelf cavities.*'

Lines 269-273: One key point this discussion is missing: the Amundsen sector ice shelves have much smaller area, so even with very high melt rates they cannot contribute much to total mass loss compared to the large cold-cavity ice shelves becoming warm.

*Authors reply:* Yes, it is absolutely right. Fig. 10 has been updated to show the relative contribution of each ice shelf in REF and PERT and some text has been added:

> '*As all the shelves turned warm, the main contributors to the total ice shelf melt in PERT are mainly the one with the largest area. In REF, the West Antarctic Ice Sheet and Peninsula sectors are responsible for 49% of the total melt for only 15% of the total Antarctic ice shelf area. In PERT, this contribution falls to 20%. In contrast, the three giant cold ice shelves (Ross, Filchner-Ronne and Amery) have their relative melt increasing in PERT. The three giants together are responsible for 50% of the total melt in PERT versus 23% in REF, for 65% of the total area (Fig. 11).*'

Line 275: Does the refreezing weaken over time, with a view to eventually disappearing? Or does the refreezing increase as melt rates increase?

*Authors reply:* This is a complex picture. The refreezing decrease rapidly in the first decade and then stabilized for 2 more decades at about 7 Gt/y before ramping up to 25 Gt/y in 2 decades and then stabilized to this value. In REF, the refreezing is about 60 Gt/y. By looking at this, we figured out that despite what the picture suggests, still experiment some refreezing (0.9 Gt/y). All the details have not been added in the new text but key elements have been added: '*... Despite this strong warming, the Ross ice shelf still exhibits a stable amount of refreezing (1 Gt yr$^{-1}$) after the first decade (not shown). [...] The Filchner-Ronne Ice Shelf exhibits a distinct behaviour: melt rates increase steadily during 100 years, but there is still a stable and significant amount of refreezing (26 Gt yr$^{-1}$) in the last decades of our experiment (Fig. 10).*'

Lines 276-277: The results from Naughten et al. (2021) are more similar than the authors imply; both studies simulate a factor of ~20 increase in FRIS mass loss, although their absolute values (both initial and final) differ.

*Authors reply:* We thank the reviewer for noticing this. However, we are not convinced that the common factor of 20 increase in mass loss is relevant. Instead, we insisted on why the simulated melt rate are different between the studies (see reply to one of the previous comments L266-268).

Line 288: Again, the word "typical" seems inappropriate here.

*Authors reply:* DONE. A full search on 'typical' has been made to correct it where necessary.

**Technical comments**
Line 62: typo in Northern
*Authors reply:* DONE

Line 211: typo in 2005
*Authors reply:* DONE

Line 213: typo in Dotson
*Authors reply:* DONE

---

## Author Comment (AC2)

**In blue: reviewer comments**
**In black: authors reply**

**General comments**
This paper, by Pierre Mathiot and Nicolas Jourdain, uses a ¼° version of NEMO, with various alterations compared with previous versions of the same model to improve the present-day climatology, forced by surface forcing from a high-end scenario in the late 23rd century. The goal of this study is to understand how ocean conditions change under such extreme forcing and the likely impact on ice shelf melt rate, which is likely to ultimately impact ice sheet mass loss and sea level rise. The model and experiments are thoroughly described and the results are interesting, although the authors correctly note that this is a highly idealised scenario and the model lacks key components of the earth system response (principally the lack of interactive ice sheets and ice shelves) that would be expected to alter the ocean conditions. I have the following suggestions to improve the manuscript:

*Authors reply:* We thank the reviewer for their positive comments on the overall paper. These in-depth comments have allowed us to clarify many aspects of our manuscript and have prompted us to go further into the description of the mechanisms.

Section 2.1: As noted on line 136, other configurations of NEMO and similar models at ¼° are unrealistic. Therefore, it would be useful to have a summary of the key changes made in this configuration that led to the improved climatology. I understand that the authors are unlikely to know the impact of every change, but it would be useful to have some discussion of the changes that are likely to have made a large improvement to the representation of the present-day conditions.

*Authors reply:* First, we have tried to better explain the motivation for the main changes in our revised section 2.1 (maximum sea ice fraction, lateral and bottom boundary conditions, increased vertical resolution, iceberg line over Bear Ridge, and tidal velocity in the three-equation melt rate). Second, we already had two sentences in the conclusion highlighting what we consider as the most important changes, so we have not added further comments:

> *'Thanks to a preliminary tuning of the sea ice model parameters and of the lateral and bottom boundary conditions at the northern end of the Antarctic Peninsula, we simulate realistic water masses in the Southern Ocean and on the Antarctic continental shelf. This is important as the performance of previous versions of eORCA025 was not good enough to be used in ocean--ice-sheet simulations (Smith et al. 2021).'*

L52-54: Please justify the reasons for thinning Getz ice shelf: why this ice shelf and not others, and how could this approach be justified? How big a difference did this single change make compared with other tunable parameters in the model?

*Authors reply:* We have added more explanations in section 2.1:

> *'After preliminary tests, the Getz ice shelf draft was artificially thinned by 200 m (keeping the grounding line unchanged) in order to compensate a longstanding bias in the thermocline depth (previously reported by Mathiot et al., 2017). The later was driving very excessive release of meltwater, which was strongly deteriorating the mean state of the Ross Sea (a connection previously described in Nakayama et al., 2020). More details on the impact of such correction are provided in Section 3.3.'*

Then in section 3.3:

> *'The total melt underneath Getz was strongly overestimated in preliminary simulations, reaching 400-500 Gt yr$^{-1}$ (not shown). By reducing the ice shelf draft of Getz (section 2), we have artificially displaced it into the model cold mixed layer, which gives more realistic melt rates. This empirical correction of the ice shelf draft is*

*nonetheless slightly too strong because it was done prior to the completion of parameter tuning.'*

Then in section 4.3:

*'The melt increase is particularly strong for Abbot and Getz ice shelves (Fig. 11) because a large portion of the ice draft is currently located in the cold winter mixed layer (Cochran et al., 2014; Wei et al., 2020) and experiences a shift to much warmer conditions in the perturbed experiment. Given that the position of the thermocline with respect to the ice draft largely drives the transition to a high melting regime, we believe that the ice draft correction applied to Getz has made the transition more realistic.'*

It is also worth noting that the shape of Getz ice shelf is not perfectly known, with uncertainties on the ice draft of approximately 100 m in BedMachine-Antarctica, but a part of the issue is also clearly related to a bias in our thermocline depth.

**L62: What is the motivation for (and impact of) applying a no-slip condition around the islands near the Antarctic Peninsula?**

*Authors reply:* The text has been slightly changed to address this comment:

- Changes in Section 2: *'A free-slip lateral boundary condition on momentum is applied with no slip condition applied locally at Bering Strait, in the whole Mediterranean sea, along the West Greenland coast and around the south Shetland, Elephant and south Orkney islands (at the Northern end of the Antarctic Peninsula). This technique is a crude method to take into account the locally complex sub-grid scale bathymetry, and it affects water mass properties as explained in section 3.2.'*

- Changes in section 3.2: *'The north end of the Antarctic Peninsula also exhibits a cold bias in REF. Preliminary analyses during the tuning process suggested that this bias was sensitive to the HSSW properties (worse when HSSW was not dense enough), to the treatment of the bathymetry, and to the lateral slip condition and bottom friction at the tip of Peninsula.'*

**L82-83: Please clarify this sentence about the calving pattern: I don't understand what this means, nor its significance, and I expect many readers will also be confused here.**

*Authors reply:* The text has been changed to:

*'The total calving rate of individual ice shelves is derived from Rignot et al. (2013) who assumed steady ice shelf fronts. As we have no information on the geographical distribution of calving for a given ice shelf, the local calving rate of each ocean cell along the front of an ice shelf is defined randomly at the beginning of the simulation preserving the total amount of calving per ice shelf. The calving rate is kept unchanged throughout the simulation.'*

**L95-96: Surface runoff from Antarctica could become regionally important in such a high-end future scenario, which should be noted here.**

*Authors reply:* We agree but we have not added anything as this was already mentioned in the next section:

*'... and that runoff from ice melting at the surface will remain zero. All these assumptions are unrealistic even for projections to 2100 (Seroussi et al., 2020; Kittel et al., 2021).'*

**L96-98: Why is this freshwater flux correction needed? What other errors in the model is this compensating for? Might these issues undermine the realism of the future scenario?**

*Authors reply:* We thank the two reviewers for pointing this out and we have added more details in the revised manuscript:

> *'On top of other freshwater fluxes (precipitation, runoff ...), a common practice in forced ocean models is to use some form of sea surface salinity restoring. This restoring is required because of the missing atmospheric feedbacks on humidity in forced models (for more details see Griffies et al., 2016). To make the model sensitivity analysis more robust, this corrective term was diagnosed from sea surface salinity restoring towards WOA2018 over the period 1999-2018 in a former simulation (the "REALISTIC" simulation described in Burgard et al., 2022) and applied as an additional climatological monthly freshwater flux in all our simulations.'*

L111-112: Although the anomaly method will correct a part of the model biases, I find it hard to believe that model biases will not affect the projected changes, especially under such an extreme scenario. Therefore, it would be useful to briefly summarise the performance of the IPSL-CM6A-LR model in this region, and consider whether there are processes that are poorly represented that may affect the realism of the surface forcing projections.

*Authors reply:* More details on IPSL-CM6 have been added to the text about the performance of IPSL-CM6A-LR:

> *'IPSL-CM6A-LR is one of the few CMIP6 models extending their scenario-based projections to 2300. In present-day conditions, IPSL-CM6-LR is cold biased by a few degrees at the surface of the Antarctic Ice Sheet (Boucher et al., 2020). On the ocean side, bottom water formation on Antarctic shelves is reasonably well represented as well as the presences of the cold and warm shelves in IPSL-CM6 (Heuzé et al. 2020; Purich et al., 2021). Sea ice extent is within the observational uncertainty in summer and slightly overestimated in winter (Boucher et al., 2020). These elements give confidence that the overall atmospheric forcings of IPSL-CM6-LR can be used to drive an ocean model'.*

With regards to the stationarity of model biases, we already had this sentence, in which we have only expanded the scenarios analyzed by Krinner et al.:

> *'Our method is expected to correct a part of the CMIP model biases that are largely stationary even under strong climate changes (as shown by Krinner et al., 2018, from preindustrial to 4xCO2)'.*

L190-191: Clarify this: are you suggesting that the updated Thwaites ice draft should improve the simulated melt rates? Or that it might have inadvertently caused larger biases?

*Authors reply:* Thwaites area changed in the recent year. Rignot et al. (2013) used a pre-2012 area (5,499 km$^2$) compared to the Paolo et al. (2023) who used a more recent area (3,116 km$^2$). We have reformulated this sentence:

> *'For Thwaites, it should be noticed that we use a recent ice shelf draft in NEMO (Morlighem et al. 2020a, 2020b) with a significantly reduced area compared to the period covered by Rignot et al. (2013), which logically decreases the integrated melt'.*

L191-192: Please clarify this statement. The ice shelf draft was reduced by 200 m to counteract a longstanding bias that was producing excessive melt rates (section 2). Here, it would be useful to repeat that the ice shelf draft was reduced by 200 m. Furthermore, it would be useful to qualify the statement that this correction was too strong: I assume that the melt rates agree better with observations than before the correction was made? Is the implication that future studies using this model could apply a similar but smaller change to the Getz ice shelf draft?

*Authors reply:* See our response to the previous comment on L52-54 and associated modifications in our manuscript. Future studies should indeed probably apply a slightly smaller correction.

 Could you give some indication of the likely reasons why the gyres intensify and increase in extent? Is this consistent with changes in ocean surface stress curl in the projected future forcing, as suggested by Gomez-Valdivia et al. (2023)? Many readers won't have read that reference, so at least discuss this possibility.

*Authors reply:* Yes, this is consistent, and we have added:

> '*This is consistent with changes in wind stress curl due to changes in the atmospheric circulation (Fig. 1e) and sea ice loss, as previously reported in projections over the 21st century (Gómez-Valdivia et al., 2023).'.*

L230-231: Can the slow trend really be "explained by slow changes in deep ocean properties at the global scale"? 100 years seems like a short time for such global deep ocean changes to be manifest. Instead, it seems more likely that the deep changes in all these locations are generated by changes in deep water source regions driven by changes in the deep circulation around Antarctica. Perhaps just replacing "global" with "circum-Antarctic" would be better?

*Authors reply:* Kissel et al. (2008) indeed suggested a millennia timescale for NADW to propagate from Arctic to glacial Southern Ocean based on paleo evidence, and Armour et al. (2016) suggested a multi centennial time scale to explain the current delay in Southern Ocean warming. In the CMIP5 outputs, however, Sallée et al. (2013) showed a warming of CDWs in a projection until 2100. They speculated that this was more a result of vertical mixing in the Southern Ocean than a change of the CDW at their source. So, we agree, circum-Antarctic is more adapted for this simulation and we have also added the reference to Sallée et al. (2013).

Armour, K., Marshall, J., Scott, J. *et al.* Southern Ocean warming delayed by circumpolar upwelling and equatorward transport. *Nature Geosci* **9**, 549–554 (2016). https://doi-org.insu.bib.cnrs.fr/10.1038/ngeo2731

Kissel, C., Laj, C., Piotrowski, A. M., Goldstein, S. L., and Hemming, S. R. (2008), Millennial-scale propagation of Atlantic deep waters to the glacial Southern Ocean, *Paleoceanography*, 23, PA2102, doi:10.1029/2008PA001624.

L237-242: It is understandable that you can't diagnose all the mechanisms, but this discussion still feels unsatisfying and like a list of possible mechanisms. Several of these mechanisms are very region-dependent, so it would be good to split this paragraph into coherent groups of regions (as done in the final paragraph of the conclusions). The relatively uniform warming might help to diagnose the most important mechanisms. For example, the currents along the shelf break are likely to be very different, and these currents are strongly linked to the supply or blocking of CDW onto the continental shelf. Figure 8 implies that these currents have changed, but it would be useful to plot the differences. It is not clear to me that the removal of sea ice and the subsequent freshening down to 400 m and deeper would lead to warming except in regions of HSSW production. Another mechanism to consider is whether the increased melt rates directly increase the overturning circulation on the continental shelf and thus help to bring more warm water onto the shelf?

*Authors reply:* This comment has pushed us to propose a deeper description of the mechanisms at play in the perturbed simulation. We have added a new figure (Fig. 10) that we use to explain how dense and warm shelves both end up as "warm–fresh shelf" with a specific current system at the shelf break (see modified section 4.2).

L249-251: What reasons might explain this difference? Is this just an area of model uncertainty? Similar question for L 274-278

*Authors reply:* Differences in the model set up may explain these different sensitivities. A paragraph has been added:

> '*Two aspects may explain these different sensitivities. First, we simulate strong increase in melt rates near the ice shelf front because of the disappearance of the cold surface layer in*

*our simulations. In the two other studies, the surface layer is still cold and the presence of an interactive ice sheet allows the ice shelf to thin and thereby to partly remain in this cold layer. Second, our parameterisation of tide-induced mixing in the three-equation system (Jourdain et al., 2019) may have a significant effect on the Ross and Ronne-Filchner melt rates. While this parameterisation has a weak effect in present-day conditions because all the available heat is consumed anyway (Hutchinson et al., 2023), the abundance of warm water in the future may enhance its role in the largest ice shelf cavities.'*

L261-262: Presumably the increase in melt rate at Getz is over-estimated due to the artificially-thinned ice shelf draft?

*Authors reply:* See our response to the previous comment on L52-54 and associated modifications in our manuscript. Wei et al. (2020) mentioned that a large portion of Getz is above the thermocline with low melt rates because within the cold surface mixed layer. Because of the thermocline bias, we thinned Getz to reproduce this. So, with our correction, a large part of Getz is somewhat realistic because it is above the model thermocline. Without this depth correction, the change in PERT for Getz could have been much weaker because our modelled Getz in REF without correction would have been warm already.

L295: While this is a very good summary overall, it would be useful to at least speculate on how these results will impact the retreat of the ice sheats and how this might in turn influence the ocean circulation.

*Authors reply:* We mention the importance of the ice sheet feedback in the conclusion. It is very difficult to compare our melt rates to other ice sheet modelling studies, first because very few provide their basal melt rates in 2300, then because the diminution of the ice shelf area and thickness in ice sheet models alter the integrated values. For example, the 15,700 Gt yr$^{-1}$ reached in our perturbed experiment in conditions of 2300 correspond to the upper end (95$^{th}$ percentile) of the ice shelf basal mass loss in 2200 in Coulon et al. (2023, under review). But their 2200 estimate is for altered ice shelves, and it decreases after 2200 due to the increasing number of collapsed ice shelves. For this reason, we prefer not to speculate on the details of the ice sheet response and its feedback to the ocean.

Coulon, V., Klose, A. K., Kittel, C., Edwards, T., Turner, F., Winkelmann, R. and Pattyn, F. (2023). Disentangling the drivers of future Antarctic ice loss with a historically-calibrated ice-sheet model. *EGUsphere*, 2023, 1-42.

**Typos etc:**

L35, L154, L197, L234: Check parentheses around citations. In some places they should be added, in others they should be removed.
*Authors reply:* DONE

L89: "equation" should be plural (equations)
*Authors reply:* DONE

L163: Fasten -> fastened?
*Authors reply:* DONE

L176: "to the exception of" -> "with the exception of"
*Authors reply:* DONE